# Continuous Viewpoint Adaptation for Single View 3D Object Reconstruction

Seunghyun Hwang [1]   Qiang Qiu [1]

## Abstract

Single-view 3D object reconstruction presents a formidable challenge in computer vision due to the inherent limitations of information obtainable from a solitary viewpoint. Recent 3D Gaussian Splatting (3DGS) inspired approaches perform a feed-forward way of learning a neural network that predicts 3D Gaussians which compose the 3D object, given a single image. However, they often struggle with occlusions and exhibit high sensitivity to small changes in input viewpoint, leading to inconsistencies and blurry artifacts in novel view renderings. Our method leverages 3DGS and introduces a new learning scheme that continuously adapts to input viewpoints. To address inherent continuity of camera viewpoints that are represented by polar and azimuthal angles, we use Neural Ordinary Differential Equations to continuously model filter subspace of neural network, thus seamlessly embedding inductive bias of perspective distortions into its structure. By continuously adapting to view-specific features, our approach fosters view consistency in 3D reconstruction, allowing better coherency and accuracy across different angles. Experiments demonstrate that our model outperforms previous methods on multiple single-view 3D reconstruction benchmark datasets and excels in extrapolating to unseen camera angles and categories.

## 1. Introduction

Single-view 3D object reconstruction remains a fundamental challenge in computer vision due to the limited information available from a single monocular image (Shen et al., 2023; Chen et al., 2024b; Szymanowicz et al., 2024). This restricted viewpoint offers minimal visual cues, making it especially difficult to infer the full 3D structure, particularly in occluded or unseen regions. Consequently, reconstruction quality is highly sensitive to the choice of input image, as changes in viewpoint can lead to substantial variations in the output geometry. Addressing this ill-posed problem requires strong prior knowledge and learned representations to compensate for missing spatial information (Shen et al., 2023). In this setting, understanding the spatial relationship between the camera and the object is critical, as it directly influences how depth and orientation are perceived and modeled (Jang & Agapito, 2021).

In this paper, we introduce a learning-based approach that makes continuous, view-adaptive predictions for single-view 3D object reconstruction, as illustrated in Figure 1. Inspired by Gaussian Splatting (Kerbl et al., 2023) and Splatter Image (Szymanowicz et al., 2024), our framework trains a neural network to predict a mixture of 3D Gaussians in a feed-forward manner from a monocular image, enabling efficient reconstructions. Camera positions in world space are continuous by nature, defined on a spherical coordinate system. To account for this inherent continuity and capture the smooth transitions between viewpoints, we devised a continuous parameterization of our view-adaptive model. Specifically, by introducing Neural Ordinary Differential Equations (NODE) to parameterize our network, we leverage the continuous dynamics of ODEs to model network parameters that adapt to varying camera angles smoothly.

Unlike existing latent ODE frameworks (Rubanova et al., 2019; Chen et al., 2018), which typically assume a single dynamic evolution across the latent space, our method conditions the initial state of the ODE on the polar angle of the input camera position and learns the evolution of each state based on azimuthal angles. This allows our model to capture distinct ODE dynamics that adapt to varying initial conditions, effectively covering the entire spherical camera space, as illustrated in Figure 2. However, directly modeling the full space of convolutional filters with NODE is computationally and memory-intensive. Therefore, we adopt a filter subspace view to convolutional neural networks (Qiu et al., 2018; Wang et al., 2021b; 2020; 2021a), where we decompose a convolutional filter as a linear combination of a set of filter atoms and coefficients. Only the parameter-efficient filter atoms are dynamically modeled by Neural ODE, while coefficients remain consistent across views.

---

[1]Purdue University, West Lafayette. Correspondence to: Seunghyun Hwang <hwang229@purdue.edu>, Qiang Qiu <qqiu@purdue.edu>.

*Proceedings of the 43rd International Conference on Machine Learning*, Seoul, South Korea. PMLR 306, 2026. Copyright 2026 by the author(s).

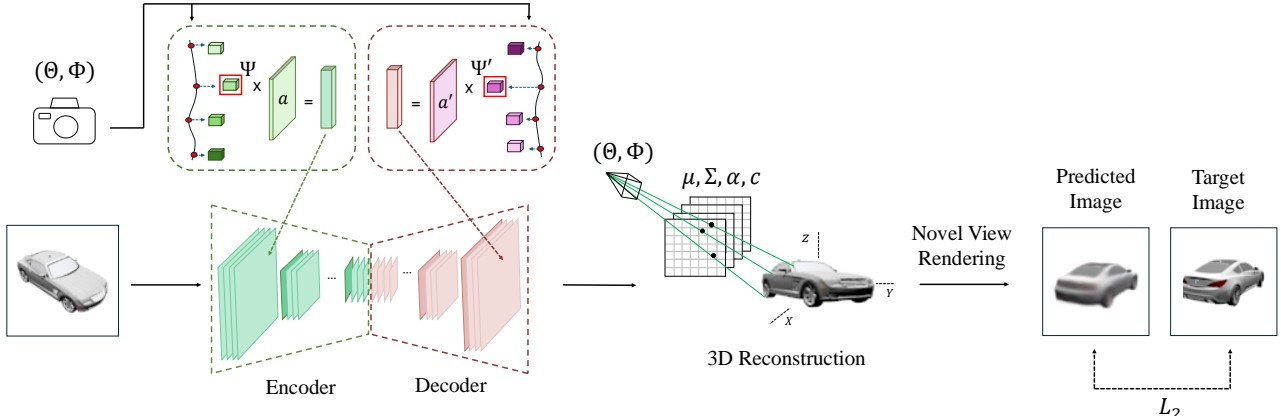

*Figure 1.* Overview of our continuous view-adaptation model for single-view 3D reconstruction. Filter atoms $\Psi$ and $\Psi'$, corresponding to the input camera view (polar angle $\Theta$ and azimuthal angle $\Phi$), are continuously generated and combined with coefficients $\mathbf{a}$ and $\mathbf{a}'$ to form the encoder and decoder layers. For easy viewing, we illustrate only the decomposition of a single convolutional filter. Each 2D image of the decoded outputs represent different parameters of mixture of 3D gaussians with each pixel serving as a container for a corresponding Gaussian. Optimization is performed by minimizing the $L_2$ distance between the renderings of the predicted and true 3D reconstructions.

We further demonstrate, both theoretically and empirically, that our continuous view-adaptive model enhances view consistency in 3D object reconstruction across varying input viewpoints. By enforcing continuity in the filter subspace, the model enables a smooth transformation from viewpoint to feature representation and ultimately to 3D output. This ensures that small changes in camera position result in gradual and coherent variations in the reconstructed shape. Such consistency is especially important in single-image 3D reconstruction, as it mitigates discontinuities when the input is given from different viewing angles. Consequently, it reduces the model's sensitivity to the choice of input viewpoint, enhancing robustness in real-world scenarios where ideal viewing angles cannot be guaranteed. (Sitzmann et al., 2019a; Tulsiani et al., 2017; Caliskan et al., 2020; Li et al., 2020).

Our approach show state-of-the-art performance on several single-view 3D object reconstruction benchmarks. Moreover, we show that our models with improved view consistency tend to generalize better to unseen viewpoints and objects. We list here our key contributions as follows:

- We propose a view adaptive model for single-view 3D object reconstruction, ensuring that depth and shape information align with the input view.

- By modeling filter subspace with NODE, we effectively capture the continuous nature of camera rotations and their impact on 3D reconstructions.

- We show that our approach facilitates view consistency in 3D reconstructions across varying input viewpoints, reducing sensitivity to input variations.

## 2. Related Works

**Single-View 3D Object Reconstruction.** In recent years, addressing the challenges in single-view 3D object reconstruction field has seen significant progress through learning-based approaches. A series of methods have leveraged Neural Radiance Fields (NeRF) to learn 3D object representations (Jang & Agapito, 2021; Yu et al., 2021; Lin et al., 2023). (Yu et al., 2021) adapts the original NeRF framework for single-view settings to infer geometry and appearance from limited observations. Moreover, series of works (Jiang et al., 2020; Cheng et al., 2023; Lin et al., 2020) employ Signed Distance Functions (SDF) (Park et al., 2019) to implicitly define object surfaces as the zero level set of a continuous scalar field, capturing fine geometric details. In addition, along with the recent advancement of generative models based on diffusion models (Ho et al., 2020), probabilistic approaches (Poole et al., 2022; Liu et al., 2023; Gu et al., 2023) are also explored to tackle this issue, with methods such as Zero123 (Liu et al., 2023) relying on pretrained large diffusion models to generate plausible 3D shapes from minimal visual input. However, these approaches often suffer from high computational costs and slow inference speeds, limiting their practical applicability.

**3D Gaussian Splatting.** To mitigate these issues, recent advancements motivated by 3D Gaussian Splatting (Kerbl et al., 2023), such as (Szymanowicz et al., 2024), introduced feed-forward frameworks that learn 3D Gaussians for efficient reconstruction, leading to considerable reductions in inference time. More recent approaches (Shen et al., 2025; Zou et al., 2024) further incorporate pretrained ViT-based encoders, e.g., DINO and DINOv2, together with transformer-based architectures to improve feature extraction for single-view 3D reconstruction. Despite these im-

provements, existing models still struggle with occluded regions, leaving room for further improvements.

## 3. Preliminaries

### 3.1. Single Image 3D Object Reconstruction with Gaussian Splatting

Gaussian Splatting (Kerbl et al., 2023) employs a mixture of 3D Gaussians to encapsulate the geometry and appearance of 3D scene or objects. Each Gaussian primitive is characterized by a set of parameters: a spatial mean $\boldsymbol{\mu} \in \mathbb{R}^3$, a covariance matrix $\boldsymbol{\Sigma} \in \mathbb{R}^{3\times3}$, an opacity value $\alpha \in \mathbb{R}$, and a view-dependent color function $c(\mathbf{v}) \in \mathbb{R}^3$.

The scene's density $\sigma(\mathbf{x})$ and color $c(\mathbf{x}, \mathbf{v})$ at any given point $\mathbf{x}$ and viewing direction $\mathbf{v}$ are computed through a weighted aggregation of these Gaussian primitives:

$$\sigma(\mathbf{x}) = \sum_i \alpha_i \mathcal{N}(\mathbf{x}; \boldsymbol{\mu}_i, \boldsymbol{\Sigma}_i), \tag{1}$$

$$c(\mathbf{x}, \mathbf{v}) = \frac{\sum_i \alpha_i \mathcal{N}(\mathbf{x}; \boldsymbol{\mu}_i, \boldsymbol{\Sigma}_i) c_i(\mathbf{v})}{\sigma(\mathbf{x})}, \tag{2}$$

where $\mathcal{N}(\mathbf{x}; \boldsymbol{\mu}, \boldsymbol{\Sigma})$ denotes the Gaussian distribution.

A notable approach (Szymanowicz et al., 2024) extends the concept of single-image 3D object reconstruction by employing a feed-forward neural network to directly predict a mixture of 3D Gaussians from a monocular image. Specifically, (Szymanowicz et al., 2024) utilizes a standard image-to-image neural network architecture to learn a function that predicts a Gaussian for each pixel of the input image. The 3D Gaussians are represented as 2D images, referred to as Splatter Images, where different parameters of Gaussians, i.e., position, covariance, opacity, and color, are stored in separate image channels. Each pixel within the Splatter Images serves as a carrier for the corresponding 3D Gaussian. The prediction network consists of an encoder $E$ that extracts image features and a decoder $D$ that generates the Gaussian parameters. Given the input image $I$, the network outputs can be written as:

$$\{\boldsymbol{\mu}_i, \boldsymbol{\Sigma}_i, \alpha_i, c_i\}_{i=1}^N = D(E(I)). \tag{3}$$

### 3.2. Neural ODE

NODE (Chen et al., 2018) offers a continuous-time perspective on neural network dynamics by conceptualizing the evolution of hidden states as a continuous process governed by a learnable differential equation, $d\mathbf{h}(t)/dt = f(\mathbf{h}(t), t, \theta)$. Here, $\mathbf{h}(t)$ is a hidden state at time $t$, $f$ is a neural network with parameters $\theta$, and $t$ represents the continuous time variable. The latent state at any arbitrary point $t_s$ can be obtained as:

$$\mathbf{h}(t_s) = \mathbf{h}(t_0) + \int_{t_0}^{t_s} f(\mathbf{h}(t), \theta) \, dt. \tag{4}$$

This integral is typically solved via numerical ODE solvers:

$$\mathbf{h}(t_s) = \text{ODESolve}(\mathbf{h}(t_0), f, (t_0, t_s), \theta), \tag{5}$$

where $t_0$ and $t_s$ denote the start and end of the integral interval, respectively.

An intriguing variant of this approach is the heavy ball Neural ODE (Xia et al., 2021), which incorporates momentum-inspired dynamics. This formulation is expressed as a second-order differential equation:

$$\frac{d^2\mathbf{h}(t)}{dt^2} + \gamma\frac{d\mathbf{h}(t)}{dt} = f(\mathbf{h}(t), t, \theta), \tag{6}$$

where $\gamma$ serves as a damping coefficient.

The Heavy Ball Neural ODE (Xia et al., 2021) offers several compelling advantages. Its momentum term facilitates rapid traversal of the loss landscape, potentially accelerating convergence. Also, the second-order dynamics enable capturing more nuanced trajectories in the feature space, potentially enhancing predictive accuracy than previous naive Neural ODE approaches (Rubanova et al., 2019).

### 3.3. Convolutional Filter Subspace

The direct modeling of high-dimensional convolutional filters in generative networks poses significant challenges in terms of computational complexity and parameter scale. To address this issue, (Qiu et al., 2018; Wang et al., 2021a;b; 2020; Chen et al., 2024a) introduced a subspace perspective on convolutional filters, inspired by the observation that these filters can be effectively approximated as linear combinations of filter atoms and coefficients.

Consider a convolutional filter $\mathbf{W} \in \mathbb{R}^{C'\times C\times K\times K}$, where $C'$ and $C$ are the number of input and output channels respectively, and $K$ is the kernel size. $\mathbf{W}$ can be decomposed over a set of $M$ filter atoms $\boldsymbol{\Psi}[i] \in \mathbb{R}^{K\times K}$ with $i \in [1, .., M]$, and atom coefficient $\mathbf{a} \in \mathbb{R}^{C'\times C\times M}$ with $1 \times 1$ filters, as $\mathbf{W} = \mathbf{a} \times \boldsymbol{\Psi}$. The filter subspace can then be expressed as $\mathbf{S} = \text{Span}\{\boldsymbol{\Psi}[1], \ldots, \boldsymbol{\Psi}[M]\}$. This introduces a paradigm where filter subspaces are model-specific, while the subspace linear combination rules, i.e., coefficients, are shared across different models (Qiu et al., 2018).

## 4. Methods

### 4.1. Continuous Modeling of View Space

We adopt the Splatter Image (Szymanowicz et al., 2024) approach introduced in Section 3.1 and learn a feed-forward

neural network that takes a single-view image as the input and output 2D images, i.e., Splatter Images, with separate image channels storing different parameters of 3D Gaussians. To improve the interpretation of depth cues from various viewpoints, we propose a view-adaptive model that utilizes distinct sets of network parameters tailored for different camera views. Camera perspectives are inherently continuous on spherical coordinates. Thus, to recognize the inherent continuity of viewpoints, we employ NODE to model the filter space in a manner that reflects this continuous nature. The guaranteed continuity NODE ensures smooth transitions within convolutional filters, maintaining continuity across views. However, directly learning and saving sets of whole network parameters is computationally and memory-intensive. Therefore, for efficient modeling, we adopt a filter subspace approach from Section 3.3, where we decompose each filer over the sets of view-specific filter atoms and cross-view shared coefficients.

We denote camera viewpoint as $(\Theta, \Phi)$, where $\Theta$ and $\Phi$ represent polar and azimuthal angle of camera position, respectively. Let $\mathbf{W}^i(\Theta, \Phi) \in \mathbb{R}^{C' \times C \times K \times K}$ be a convolutional filter in $i$-th layer of our encoder-decoder shaped feed-forward network for a given viewpoint $(\Theta, \Phi)$, where $C'$ and $C$ represent the number of input and output channels respectively, and $K$ is the kernel size. We decompose this filter over a set of $M$ filter atoms and coefficient as:

$$\mathbf{W}^i(\Theta, \Phi) = \mathbf{a}^i \mathbf{\Psi}^i(\Theta, \Phi), \quad (7)$$

where $\mathbf{\Psi}^i(\Theta, \Phi) \in \mathbb{R}^{M \times K \times K}$ are view-dependent filter atoms and $\mathbf{a}^i \in \mathbb{R}^{C' \times C \times M}$ are view-shared coefficients as 1x1 filters.

**View-Conditional Filter Atoms.** To capture the continuous nature of spherical camera coordinates, we employ a heavy ball Neural ODE (Xia et al., 2021), the current state-of-the-art NODE system, to generate view-conditional filter atoms. The evolution of atoms at every $i$-th layer of our network is governed by:

$$\frac{d^2 \mathbf{\Psi}^i(\Phi)}{d\Phi^2} + \gamma \frac{d\mathbf{\Psi}^i(\Phi)}{d\Phi} = f(\mathbf{\Psi}^i(\Phi), \Phi; \theta), \quad (8)$$

where $\Phi$ is the azimuthal angle, $\gamma$ is a damping coefficient, and $f$ is a neural network parameterized by $\theta$. The initial conditions for position and momentum are learned functions of the polar angle, $\Theta$, as:

$$\mathbf{\Psi}^i(\Phi_0) = u_{\text{pos}}(\Theta), \quad \frac{d\mathbf{\Psi}^i(\Phi)}{d\Phi(\Phi_0)} = u_{\text{vel}}(\Theta). \quad (9)$$

Here, $u_{\text{pos}}$ and $u_{\text{vel}}$ represent small MLP networks that output initial position and velocity, respectively. Note that when processed by $u_{\text{pos,vel}}$, $\Theta$ is normalized to [0, 1] from its original scale of [0°, 180°].

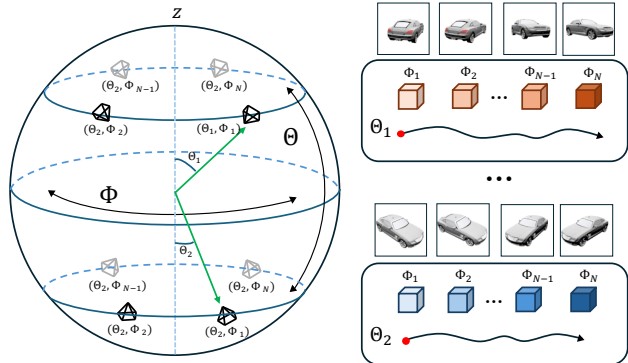

*Figure 2.* Illustration of continuous modeling of filter subspace with camera viewpoints. Camera positions are defined on a spherical world space, with polar angle $\Theta$ and azimuthal angle $\Phi$. The filter atoms are generated by ODE flow whose initial values are conditioned on $\Theta$ and evolves according to $\Phi$.

Filter atoms are sampled from continuous filter subspace modeled by $f$, as follows:

$$\mathbf{\Psi}^i(\Phi_s) = \mathbf{\Psi}^i(\Phi_0) + \int_{\Phi_0}^{\Phi_s} f(\mathbf{\Psi}^i(\Phi), \Phi; \theta) \, d\Phi. \quad (10)$$

As illustrated in Figure 2, filter atoms at any arbitrary azimuthal angle $\Phi_s$ are produced through ODE integration. The ODE system constraints the endpoint of integration interval to less than or equal to 1. Therefore, $\Phi$ of scale $[-180°, 180°]$ is normalized into [0, 1]. Accordingly, the $\Phi_0$ equals to 0 in our settings. The integral is solved numerically using ODE solvers, e.g., Runge–Kutta methods, and can be expressed as $\mathbf{\Psi}^i(\Phi_s) = \text{ODESolve}(\mathbf{\Psi}^i(0), f, (0, \Phi_s), \theta)$.

The above modeling of filter atoms allows us to model the continuous camera system in the filter subspace, instilling the network with an inductive bias about the viewspace.

**Optimization.** We decompose layers of both the encoder and decoder parts of our U-shaped (Ronneberger et al., 2015) feed-forward network into sets of view-dependent filter atoms and view-shared coefficients. To generate filter atoms with respect to input viewpoints, two separate atom generators, realized by different NODEs, are employed for encoder and decoder, respectively. Mathematically, given an input image $\mathbf{X}$ captured at view angle $(\Theta, \Phi)$, our view-adaptive encoder $\mathbf{W}_{enc}(\Theta, \Phi)$ extracts feature embeddings $\mathbf{Z}$, which are subsequently processed by the convolutional decoder $\mathbf{W}_{dec}(\Theta, \Phi)$ to generate 3D Gaussians. Each layer of encoder and decoder are decomposed as $\mathbf{W}_{enc}^i(\Theta, \Phi) = \mathbf{a}^i \mathbf{\Psi}^i(\Theta, \Phi)$ and $\mathbf{W}_{dec}^i(\Theta, \Phi) = \mathbf{a}'^i \mathbf{\Psi}'^i(\Theta, \Phi)$, respectively. All model parameters, including those of atom generators, are trained end-to-end by measuring the difference between rendering of the predicted and true 3D Gaussians using the $L_2$ loss. The overall schematic view of the learning process is shown in Figure 1.

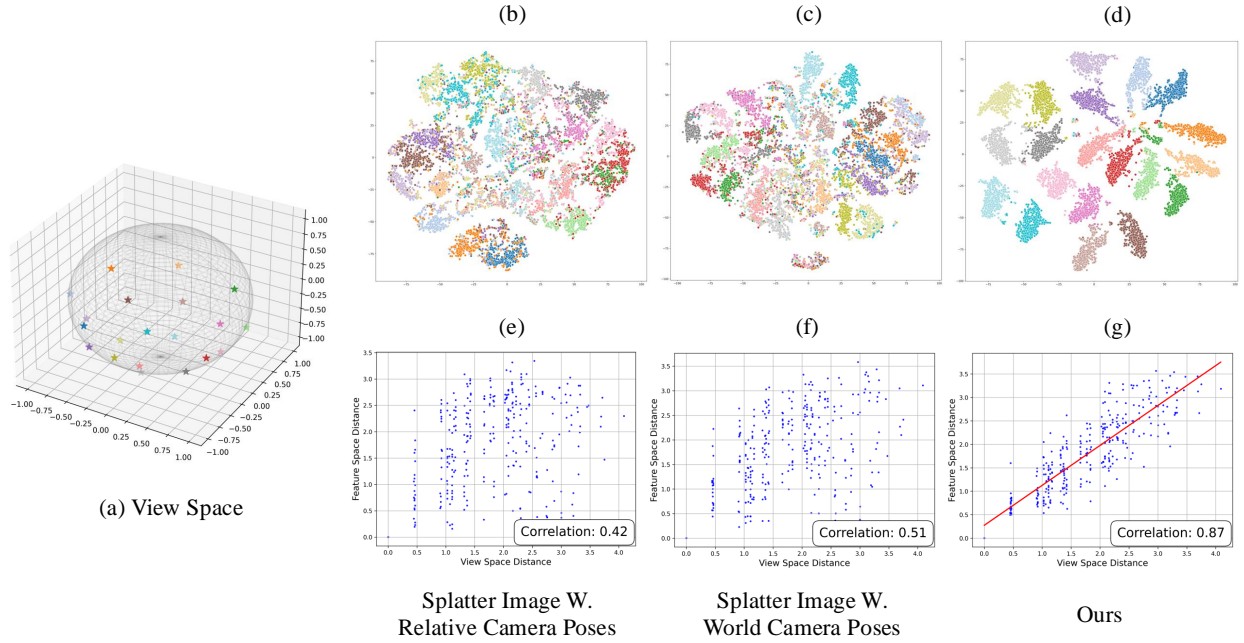

*Figure 3.* Comparison analysis of feature embeddings from Splatter Image (Szymanowicz et al., 2024) with relative camera poses, i.e., (b), (e), Splatter Image with world camera poses, i.e., (c), (f), and our model, i.e., (d), (g), on the SRN-Chairs test split. From 1317 chair objects, images at 20 distinct viewpoints are sampled and encoded by different models. (a) shows sampled camera viewpoints visualized by different colors on the spherical view space. (b), (c), and (d) present 2D t-SNE visualizations of feature embeddings generated from different models. (e), (f), and (g) show pairwise Mahalanobis distances between instances in the view space and feature space, together with their Pearson correlation scores of 0.42, 0.51, and 0.87, respectively.

### 4.2. Smooth View Adaptation and View Consistency

As the camera moves through the spherical world space, the filter atoms that constitute each layer of $\mathbf{W}_{enc}$ evolve smoothly according to a learned ODE system, effectively capturing view-dependent features. We theoretically demonstrate that small changes in input viewpoints result in small variations in the continuously generated filter atoms, which in turn lead to minimal changes in the output features.

Specifically, suppose the input at $i$-th layer of the encoder as $\mathbf{Z}^i(s)$ ($s \in \mathcal{S}, \mathcal{S} \subset \mathbb{Z}^2$), the local input norm is defined as $\|\mathbf{Z}^i\|_{2,N_s} := (\sum_{s' \in N_s} \mathbf{Z}^i(s-s')^2)^{1/2}$, and the convolution is expressed as $\langle \mathbf{Z}^i, w \rangle_{N_s} := \sum_{s' \in N_s} \mathbf{Z}^i(s-s')w(s')$, where $N_s \subset \mathcal{S}$ is a local Euclidean grid centered at $s$. The decomposed convolution at $i$-th layer can then be found as $\mathbf{Z}^{i+1}(s) = g(\sum_{j=1}^M \mathbf{a}^i[j]\langle \mathbf{Z}^i, \mathbf{\Psi}^i[j]\rangle_{N_s} + b)$, where $\mathbf{Z}^i(s)$ is the feature embeddings, $\mathbf{\Psi}^i$ and $\mathbf{a}^i$ are the corresponding filter atoms and coefficients. The $g$ is the activation function, e.g., SiLU, and $b$ is bias term.

The following proposition demonstrates that the Lipschitz continuity of the filter atoms propagates to the network outputs, as:

**Proposition 4.1** (Continuity of View-Specific Features). *Let $\mathbf{Z}_u^i$ and $\mathbf{Z}_v^i$ be inputs to the $i$-th layer from adjacent views, and $\mathbf{\Psi}_u^i$ and $\mathbf{\Psi}_v^i$ represent two continuously generated sets of filter atoms corresponding to those views, which together*

*with the common atom coefficients $\mathbf{a}^i$ form $i$-th layer of the convolutional layer. If the difference between $\mathbf{Z}_u^i$ and $\mathbf{Z}_v^i$ is small (i.e., $\|\mathbf{Z}_u^i - \mathbf{Z}_v^i\|_2 \leq \delta$) for some $\delta$, and for non-expansive activation function, e.g., SiLU, we can upper bound the changes in the corresponding outputs $\mathbf{Z}_u^{i+1}$ and $\mathbf{Z}_v^{i+1}$ in terms of the changes in the filter atoms as:*

$$\|\mathbf{Z}_u^{i+1} - \mathbf{Z}_v^{i+1}\|_2 \leq (\|\mathbf{a}^i\|_2 \lambda \sqrt{|\mathcal{S}|} + \delta) \cdot \|\mathbf{\Psi}_u^i - \mathbf{\Psi}_v^i\|_2,$$
$$with \ \lambda = \sup_{s \in \mathcal{S}} \|\mathbf{Z}_u^i\|_{2,N_s}. \tag{11}$$

Proof is provided in Appendix Section A. The Proposition 4.1 suggests that when the view difference between input images is small, the resulting feature embeddings (i.e., processed by continuously generated atoms corresponding to those viewpoints) will also exhibit small variations, with the continuity of the atoms ensured by the NODE framework. Although Proposition 4.1 is the analysis of one layer, the continuous property holds from the first to the last layer, i.e., the whole model.

**Embeddings Space Analysis.** We further empirically validate the continuous dependence relationship between feature embeddings and input viewpoints through an in-depth analysis of the embeddings-viewpoints correlation, as presented in Figure 3. The figure illustrates a comparison of feature embeddings extracted from the encoder of ours, Splatter Image (Szymanowicz et al., 2024) with relative

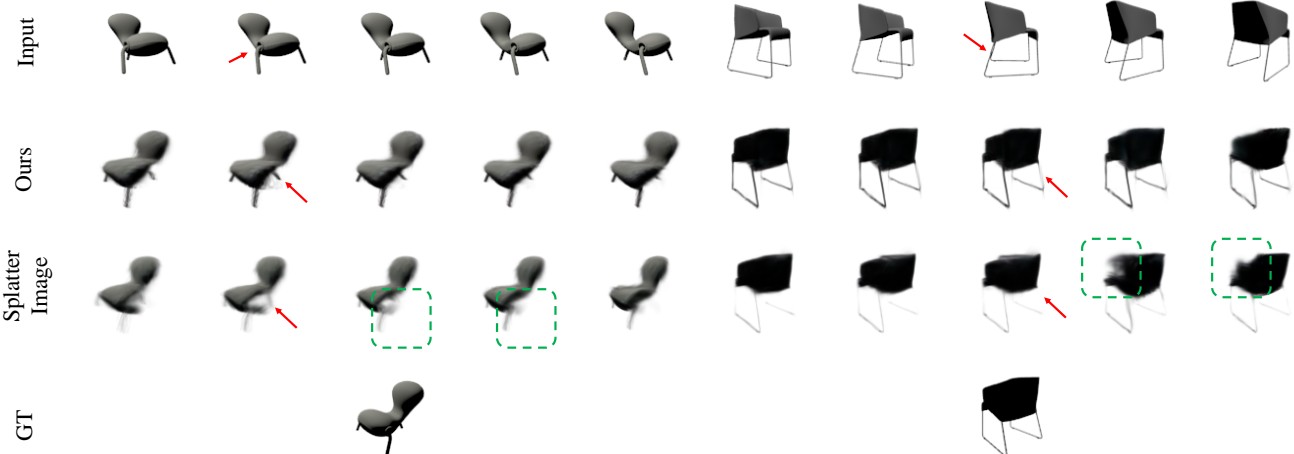

*Figure 4.* Comparison of 3D reconstruction consistency and occlusion handling. The top row shows the original single-view images with slight variations in viewing angles. Second and third row shows the results of novel view renderings from our model and the baseline Splatter Image (Szymanowicz et al., 2024). Last row shows the ground-truth rendering. The red arrow points to the occluded regions from the input view, e.g., hidden legs. The green boxes show where the baseline results fail to maintain coherence across different input views.

camera poses, and Splatter Image with world camera poses. As the vanilla Splatter Image uses relative camera poses, for fair comparison to ours, we also trained a Splatter Image with world camera positions as input. For data, we used test split of SRN-Chairs dataset, where we first sampled 20 different camera viewpoints and for each view, we encoded 1317 distinct types of chairs. Encoded features are visualized using 2-dimensional t-SNE.

Figure 3 reveals that the vanilla Splatter Image model fails to achieve clear viewpoint-specific clustering, leading to mixed feature distributions across viewpoints. Splatter Image model trained with world camera poses also exhibit limited ability to capture viewpoint-dependent features although world camera poses are given as an input. Conversely, features extracted from the learned encoder of ours demonstrate well-clustered groupings by viewpoints, indicating strong connectivity between features and viewpoints. Moreover, to show how changes in view space affect changes in features space, we measured the Pearson correlation scores for the pairwise distance between instances in both spaces. Ours showed a strongest correlation score of 0.87, which implies that proximity in view space corresponds to proximity in feature space. Conformably, we show that small viewpoint differences lead to small embedding changes.

**Viewpoint Robustness.** The continuous dependence relationship between feature embeddings and input viewpoints similarly applies to the relationship between feature embeddings and output 3D reconstructions. Specifically, the decoder, with each $i$-th layer decomposed as $\mathbf{W}_{dec}^i(\Theta, \Phi) = \mathbf{a}'^i \mathbf{\Psi}'^i(\Theta, \Phi)$, maps feature embeddings $\mathbf{Z}^i$ to outputs $\mathbf{Z}^{i+1}$. Here, the final output is represented as a set of 2D images, i.e., Splatter Images, where each image holds parameters of the 3D Gaussians that compose the object. Given the feature embeddings $\mathbf{Z}^i(s)(s \in \mathcal{S}, \mathcal{S} \subset \mathbb{Z}^2)$, the

decomposed convolution at $i$-th layer of decoder can then be found as $\mathbf{Z}^{i+1}(s) = g(\sum_{j=1}^M \mathbf{a}'^i[j]\langle \mathbf{Z}^i, \mathbf{\Psi}'^i[j]\rangle_{N_s} + b)$.

**Remark 4.2.** *According to Proposition 4.1, the difference between feature embeddings at each layer is upper bounded by the difference in the continuously generated filter atoms, for some small difference of $\delta$ between inputs from close views. Similarly, for the decoder, where each layer is also decomposed over continuously generated view-dependent filter atoms and cross-view shared coefficients, if the difference between the input features at $i$-th layer is small (i.e., $\|\mathbf{Z}_u^i - \mathbf{Z}_v^i\|_2 \leq \delta'$), we can upper bound the variations in the corresponding outputs in terms of the changes in the filter atoms.*

Therefore, the Lipschitz continuity of decoder atoms propagates to the final 3D reconstruction. Overall, the continuity from viewpoint to features and from features to 3D outputs ensures that minor input perturbations lead to smooth, bounded changes in the reconstructed 3D shape. This prevents abrupt changes or inconsistencies in the 3D reconstructions when different input viewpoints are given, increasing robustness to minor viewpoint shifts.

Figure 4 illustrates how our model improves robustness across varying input viewpoints by maintaining reconstruction accuracy across different inputs. Novel view renderings under slight rotations show our method consistently produces coherent outputs, while the baseline (Szymanowicz et al., 2024) introduces artifacts and inconsistencies. Our model also reconstructs hidden parts, e.g., occluded legs, more reliably, leveraging adjacent view understanding. In contrast, the baseline often fails to recover such details, highlighting the limitations of models with weak view consistency in realistic, unconstrained settings. Visual comparisons of opacity maps featured by the baseline and

| Method | 1-view Cars | | | 1-view Chairs | | |
|---|---|---|---|---|---|---|
| | PSNR↑ | SSIM↑ | LPIPS↓ | PSNR↑ | SSIM↑ | LPIPS↓ |
| SRN (Chang et al., 2015) | 22.25 | 0.88 | 0.129 | 22.89 | 0.89 | 0.104 |
| CodeNeRF (Jang & Agapito, 2021) | 23.80 | 0.91 | 0.128 | 23.66 | 0.90 | 0.166 |
| FE-NVS (Guo et al., 2022) | 22.83 | 0.91 | 0.099 | 23.21 | 0.92 | 0.077 |
| ViewestDiff w/o D (Szymanowicz et al., 2023) | 23.21 | 0.90 | 0.116 | 24.16 | 0.91 | 0.088 |
| PixelNeRF (Yu et al., 2021) | 23.17 | 0.89 | 0.146 | 23.72 | 0.90 | 0.128 |
| VisionNeRF (Lin et al., 2023) | 22.88 | 0.90 | 0.084 | 24.48 | 0.92 | 0.077 |
| NeRFDiff w/o NGD (Gu et al., 2023) | 23.95 | 0.92 | 0.092 | **24.80** | **0.93** | 0.070 |
| Splatter Image (Szymanowicz et al., 2024) | 24.00 | 0.92 | 0.078 | 24.43 | **0.93** | **0.067** |
| Ours | **24.46** | **0.93** | **0.073** | 24.76 | **0.93** | 0.071 |

*Table 1.* Comparison of different methods on 1-view Cars and Chairs datasets. PSNR, SSIM, and LPIPS are reported for both categories. Top four methods use absolute camera poses, while the bottom four comparison methods use relative camera poses.

ours are showed in the Figure 10 in the Appendix Section C, to highlight how occluded parts are better captured in our approach. We further provide quantitative comparisons across multiple conditioning viewpoints on the ShapeNet-Cars and ShapeNet-Chairs datasets in Table 9, as well as on the Google Scanned Objects (GSO) dataset (Downs et al., 2022) and the Teddybears and Hydrants categories from CO3D (Reizenstein et al., 2021) in Table 10, presented in Appendix Section C.

## 5. Experiments

We conduct extensive experiments to evaluate our method's performance in single-view 3D object reconstruction tasks across various settings, including standard benchmarks and robustness testings. Robustness testings include testing on unseen views and unseen categories. For unseen views, we introduce view-extrapolation and view-interpolation settings, where we test on unseen views using filter atom interpolation. For unseen category testing, we test on totally different object categories unseen during the training.

**Implementations.** Following (Szymanowicz et al., 2024), we adopted U-Net (Ronneberger et al., 2015) shaped architecture from (Song et al., 2020) as our prediction model, and adopted three 1x1 convolutional layers with activation functions to represent the NODE function. 3-layer MLP is used for generating initial velocity and positions of the NODE. More details are provided in Appendix Section B.1.

**Datasets.** We tested our models on seven different 3D object reconstruction benchmark datasets. For synthetic datasets, we used Cars and Chairs from ShapeNet-SRN dataset (Sitzmann et al., 2019b). For multi-category validation, we used multi-category ShapeNet datasets from (Kato et al., 2018). For validation on real images, we used teddybears and hydrants categories from CO3D (Reizenstein et al., 2021) dataset. To further demonstrate scalability

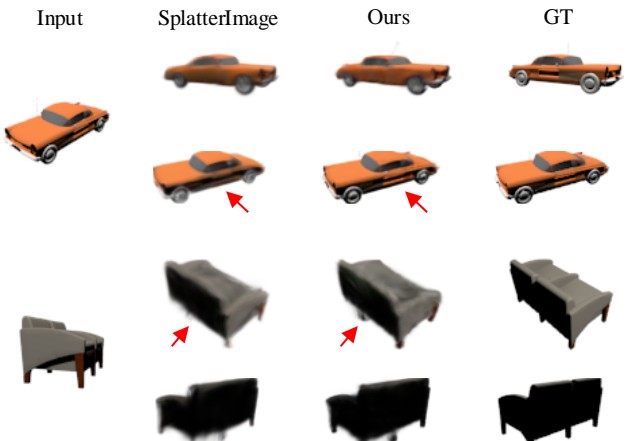

*Figure 5.* Visual comparison between our approach and the Splatter Image on ShapeNet-Cars and Chairs dataset. Ours show more accurate results especially in the regions pointed by the red arrow.

and generalization on large scale datasets, we trained our model on Objaverse (Deitke et al., 2023) which contains over 1k object categories, and tested on Google Scanned Objects (Downs et al., 2022). Further details about data preprocessings are provided in the Appendix Section B.2.

**Evaluation.** We evaluate the models by measuring their performance on novel view renderings using three complementary metrics. First, we use Peak Signal-to-Noise Ratio, i.e., PSNR, to measure the signal fidelity of the rendered images. Next, we employ Learned Perceptual Image Patch Similarity i.e., LPIPS, to evaluate perceptual similarity, leveraging a pretrained VGG network. Finally, we use Structural Similarity Index i.e., SSIM, to assess the perceived image quality of the rendered images.

### 5.1. Single-view 3D Reconstructions

Here, we present comparison results for single-view 3d reconstructions on ShapeNet Cars and Chairs, multi-category

| Dataset | Method | PSNR↑ | SSIM↑ | LPIPS↓ |
|---|---|---|---|---|
| | PixelNeRF | 26.80 | 0.91 | 0.108 |
| | FE-NVS | 27.08 | 0.92 | 0.082 |
| NMR | VisionNeRF | 28.76 | 0.93 | 0.065 |
| | Splatter Image | 29.38 | 0.95 | **0.047** |
| | **Ours** | **29.51** | **0.954** | **0.047** |

*Table 2.* Comparison on the NMR dataset. Our method achieves state-of-the-art performance for single-view 3D reconstruction on the multi-category ShapeNet benchmark.

| Object | Method | PSNR↑ | SSIM↑ | LPIPS↓ |
|---|---|---|---|---|
| | PixelNeRF | 21.76 | 0.78 | 0.207 |
| Hydrants | Splatter Image | 21.80 | 0.80 | 0.150 |
| | **Ours** | **21.81** | **0.82** | **0.147** |
| | PixelNeRF | 19.38 | 0.65 | 0.290 |
| Teddybears | Splatter Image | 19.44 | **0.73** | 0.231 |
| | **Ours** | **19.50** | 0.72 | **0.230** |

*Table 3.* Comparison of different methods on CO3D Hydrants and Teddybears. PSNR, SSIM, and LPIPS are reported for each method.

ShapeNet, and CO3D Teddybears and Hydrants datasets.

**ShapeNet-Cars&Chairs.** The quantitative results on the ShapeNet-Cars and Chairs are presented in Table 1. We achieve best scores on the ShapeNet-Cars for all three metrics, while maintaining comparable scores on the ShapeNet-Chairs. The NeRFDiff (Gu et al., 2023) model which presents best PSNR score relies on a prior knowledge from conditional diffusion network. Besides NeRFDiff, we obtained highest PSNR score. Qualitative comparison is presented in Figure 5 with more visual examples in Figure 11 in the Appendix Section C.

**Multi-Category ShapeNet.** We outperform all other prior works on the multi-category ShapeNet dataset, demonstrating our method's strong generalization capabilities in multi-category scenarios, as shown in Table 2. Qualitative results are presented in Figure 13 in the Appendix Section C.

**CO3D.** Our approach outperforms others on the CO3D Hydrants dataset across all metrics and achieves superior PSNR

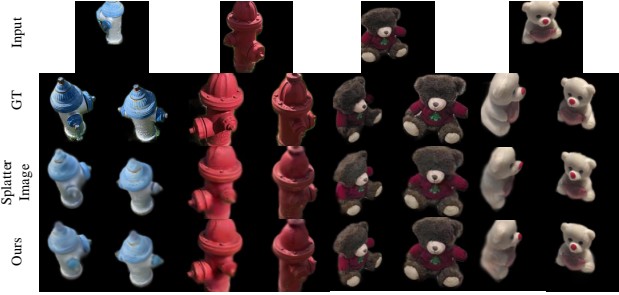

*Figure 6.* Visual comparison between our approach and the Splatter Image on CO3D Hydrants and Teddybears datasets.

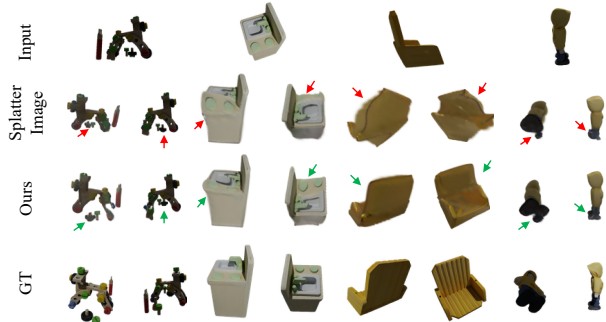

*Figure 7.* Qualitative comparison on Google Scanned Objects. The first row shows the input image, followed by novel-view renderings from Splatter Image and our method, respectively. The final row shows the ground-truth novel views. Our method shows more consistent appearance across viewpoints as pointed by the green arrows.

and LPIPS scores for the Teddybears category, demonstrating its effectiveness on real-world object datasets, as shown in Table 3. Visual comparisons are presented in Figure 6.

**Objaverse-LVIS and Google Scanned Objects.** We compare our methods with Splatter Image (Szymanowicz et al., 2024) and OpenLRM, an open-source version of the LRM (Hong et al., 2024). Our method consistently outperforms both works across all metrics, demonstrating strong robustness and generalization across diverse object categories and structures, as shown in Table 4. Qualitative comparisons are shown in Figure 7, where our method produces more consistent novel-view renderings with improved reconstruction quality.

| Dataset | Method | PSNR↑ | SSIM↑ | LPIPS↓ |
|---|---|---|---|---|
| | OpenLRM | 18.06 | 0.84 | 0.129 |
| GSO | Splatter Image | 21.06 | 0.88 | 0.110 |
| | **Ours** | **21.74** | **0.89** | **0.090** |

*Table 4.* Evaluation on the Google Scanned Objects dataset. Our method demonstrates improved generalization performance across unseen real-world object categories.

### 5.2. Robustness Evaluation

**Unseen View Generalization.** We evaluate our model's generalization to unseen camera poses using two different subsets of viewpoints not encountered during training. The filter atoms at these unseen view angles can be either interpolated or extrapolated based on the target viewpoint using NODE, which is expected to improve the model's understanding of novel angles. Illustration of the two separate view splits are presented in Figure 9 in the Appendix Section B.2. Table 5 shows that we outperform the vanilla Splatter Image model across all object categories for both scenarios. Qualitative comparisons are presented in Figure 8.

| | | Interpolation | | | Extrapolation | | |
|---|---|---|---|---|---|---|---|
| Object | Method | PSNR↑ | SSIM↑ | LPIPS↓ | PSNR↑ | SSIM↑ | LPIPS↓ |
| Chairs | Splatter Image | 21.59 | 0.89 | 0.141 | 20.82 | 0.87 | 0.160 |
| | **Ours** | **23.51** | **0.92** | **0.101** | **21.48** | **0.89** | **0.136** |
| Cars | Splatter Image | 22.74 | 0.90 | 0.131 | 22.14 | 0.89 | 0.130 |
| | **Ours** | **23.21** | **0.91** | **0.120** | **22.59** | **0.90** | **0.123** |

*Table 5.* Comparison of different methods under view extrapolation and interpolation scenarios for ShapeNet-Cars and Chairs.

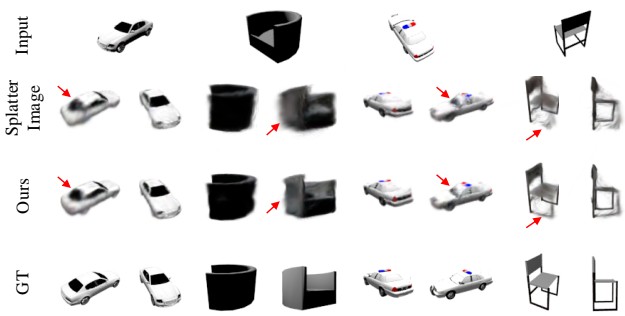

*Figure 8.* Qualitative comparison between our approach and the Splatter Image under view interpolation (i.e., first and second columns) and view extrapolation (i.e., third and fourth columns) scenarios for ShapeNet-Cars and Chairs dataset. Our approach is better handling at occluded regions pointed by the red arrow.

**Unseen Category Generalization.** To assess cross-category generalization, we test the models trained on ShapeNet-Cars on ShapeNet-Chairs, and vice-versa. With cross-category evaluation, we show how our approach with view-aware filter atoms can better generalize to unseen categories with learned geometric priors. Our approach demonstrates superior generalization capabilities compared to the vanilla Splatter Image as presented in Table 6. Visual results are presented in Figure 12 in the Appendix Section C.

| Trainset Category | Method | PSNR↑ | SSIM↑ | LPIPS↓ |
|---|---|---|---|---|
| ShapeNet-Cars | Splatter Image | 14.49 | 0.79 | 0.23 |
| | **Ours** | **15.24** | **0.80** | **0.21** |
| ShapeNet-Chairs | Splatter Image | 16.45 | 0.800 | 0.22 |
| | **Ours** | **16.46** | **0.802** | **0.21** |

*Table 6.* Evaluation results on unseen category reconstruction task. Models trained on ShapeNet Chairs are tested on multi-category ShapeNet dataset, and vice-versa.

**Input Viewpoint Perturbations.** To evaluate the model reliance on accurate camera poses and its robustness to noisy viewpoint inputs, we perturb both azimuth and polar angles during inference. Specifically, we introduce noise levels of 5%, 10%, and 15% to the input camera parameters. On the CO3D Teddybears dataset, PSNR decreases from 18.55 to 18.04 and 17.74 as the perturbation level increases. Similarly, on the CO3D Hydrants dataset, PSNR decreases from 21.62 to 21.57 and 21.28 under the same perturbations. These results show that the model remains stable under

moderate pose noise, with gradual degradation as noise increases. This indicates that the method is reasonably robust to imperfect camera pose inputs while maintaining consistent reconstruction quality.

### 5.3. Ablations

To analyze the impact of individual components in our framework, we perform ablation studies, using shorter training steps than models in Table 1, on the ShapeNet Cars dataset, as shown in Table 7. Starting with a baseline vanilla Splatter Image, we incrementally add features and evaluate their effects using shorter training iterations for efficiency. First, we incorporate world space camera poses to confirm the benefits of using camera information. Next, we validate conditioning filter atoms on camera views by decomposing the network into filter atoms and coefficients, employing a simple 3-layer MLP as atom generator, instead of NODE. We then test the effectiveness of using NODE for continuous view space modeling, initially conditioning only on the azimuthal angle Φ. Finally, we demonstrate the improvement gained by conditioning NODE on both the polar angle Θ and the azimuthal angle Φ. As summarized in Table 7, each addition enhances performance, with our full model in the last row achieving the best results across all metrics.

| | PSNR ↑ | SSIM ↑ | LPIPS ↓ |
|---|---|---|---|
| Baseline | 21.81 | 0.88 | 0.166 |
| + Camera | 22.13 | 0.88 | 0.156 |
| + DCF | 22.19 | 0.88 | 0.155 |
| + ODE($\Phi$) | 22.64 | 0.89 | 0.143 |
| + ODE($\Phi, \Theta$) | **23.12** | **0.90** | **0.129** |

*Table 7.* Ablation studies by adding different components.

## 6. Limitations and Conclusion

This work focuses on reconstructing single 3D objects and does not directly generalize to complex 3D scene reconstruction. Extending our approach to full scenes with multiple interacting objects, occlusions, and background clutter remains an important direction for future work. Additionally, our method relies on known camera viewpoints to generate view-dependent filter atoms. While common in controlled datasets, this requirement may limit applications where view annotations are unavailable or imprecise. Despite these limitations, we presented a novel approach for single-view 3D object reconstruction using a continuous, view-adaptive framework that enforces view consistent reconstructions with respect to different input selections. By utilizing neural ODEs to model filter subspaces conditioned on continuous camera angles, we ensure that small input variations result in bounded changes in the reconstructed shape, reducing reliance on a specific input image and advancing the robustness of single-view 3D reconstruction.

## Impact Statement

This paper presents work whose goal is to advance the field of Machine Learning. There are many potential societal consequences of our work, none which we feel must be specifically highlighted here.

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

# A. Proof of Theorem 2

**Decomposed Convolution.** Recall the decomposed convolution at $i$-th layer can be expressed as $\mathbf{Z}^{i+1}(s) = g(\sum_{j=1}^{M} \mathbf{a}^i[j]\langle \mathbf{Z}^i, \boldsymbol{\Psi}^i[j]\rangle_{N_s} + b)$, where $\mathbf{Z}^i(s)$ is the feature embeddings, $\boldsymbol{\Psi}^i$ and $\mathbf{a}^i$ are the corresponding filter atoms and coefficients. The $g$ is the activation function, e.g., SiLU, and $b$ is bias term.

**Proposition A.1** (Continuity of View-Specific Features). *Let $\mathbf{Z}_u^i$ and $\mathbf{Z}_v^i$ be inputs to the $i$-th layer from adjacent views, and $\boldsymbol{\Psi}_u^i$ and $\boldsymbol{\Psi}_v^i$ represent two continuously generated sets of filter atoms corresponding to those views, which together with the common atom coefficients $\mathbf{a}^i$ form $i$-th layer of the convolutional layer. If the difference between $\mathbf{Z}_u^i$ and $\mathbf{Z}_v^i$ is small (i.e., $\|\mathbf{Z}_u^i - \mathbf{Z}_v^i\|_2 \leq \delta$) for some $\delta$, and for non-expansive activation function, e.g., SiLU, we can upper bound the changes in the corresponding outputs $\mathbf{Z}_u^{i+1}$ and $\mathbf{Z}_v^{i+1}$ in terms of the changes in the filter atoms as:*

$$\|\mathbf{Z}_u^{i+1} - \mathbf{Z}_v^{i+1}\|_2 \leq (\|\mathbf{a}^i\|_2 \lambda \sqrt{|\mathcal{S}|} + \delta) \cdot \|\boldsymbol{\Psi}_u^i - \boldsymbol{\Psi}_v^i\|_2,$$
$$\text{with } \lambda = \sup_{s \in \mathcal{S}} \|\mathbf{Z}_u^i\|_{2,N_s}. \tag{12}$$

*Proof.* Since $g$ is non-expansive, for all $s$,

$$\left|\mathbf{Z}_u^{i+1}(s) - \mathbf{Z}_v^{i+1}(s)\right|$$
$$\leq \left|\sum_{j=1}^{M} \mathbf{a}^i[j]\langle \mathbf{Z}_u^i, \boldsymbol{\Psi}_u^i[j]\rangle_{N_s} - \sum_{j=1}^{M} \mathbf{a}^i[j]\langle \mathbf{Z}_v^i, \boldsymbol{\Psi}_v^i[j]\rangle_{N_s}\right|$$
$$\leq \|\mathbf{a}^i\|_2 \left(\sum_{j=1}^{M}\left|\langle \mathbf{Z}_u^i, \boldsymbol{\Psi}_u^i[j]\rangle_{N_s} - \langle \mathbf{Z}_v^i, \boldsymbol{\Psi}_v^i[j]\rangle_{N_s}\right|^2\right)^{1/2}. \tag{13}$$

Using the property of inner products, we decompose

$$\langle \mathbf{Z}_u^i, \boldsymbol{\Psi}_u^i[j]\rangle_{N_s} - \langle \mathbf{Z}_v^i, \boldsymbol{\Psi}_v^i[j]\rangle_{N_s}$$
$$= \langle \mathbf{Z}_u^i - \mathbf{Z}_v^i, \boldsymbol{\Psi}_u^i[j]\rangle_{N_s} + \langle \mathbf{Z}_v^i, \boldsymbol{\Psi}_u^i[j] - \boldsymbol{\Psi}_v^i[j]\rangle_{N_s}. \tag{14}$$

Applying the triangle inequality,

$$\left|\langle \mathbf{Z}_u^i, \boldsymbol{\Psi}_u^i[j]\rangle_{N_s} - \langle \mathbf{Z}_v^i, \boldsymbol{\Psi}_v^i[j]\rangle_{N_s}\right|$$
$$\leq \left|\langle \mathbf{Z}_u^i - \mathbf{Z}_v^i, \boldsymbol{\Psi}_u^i[j]\rangle_{N_s}\right| + \left|\langle \mathbf{Z}_v^i, \boldsymbol{\Psi}_u^i[j] - \boldsymbol{\Psi}_v^i[j]\rangle_{N_s}\right|. \tag{15}$$

Each term in Eq.15 can be bounded using Cauchy-Schwarz inequality. Using the assumption $\|\mathbf{Z}_u^i - \mathbf{Z}_v^i\|_2 \leq \delta$, the first term can be bounded as

$$\left|\langle \mathbf{Z}_u^i - \mathbf{Z}_v^i, \boldsymbol{\Psi}_u^i[j]\rangle_{N_s}\right| \leq \|\mathbf{Z}_u^i - \mathbf{Z}_v^i\|_{2,N_s} \cdot \|\boldsymbol{\Psi}_u^i[j]\|_{2,N_s}$$
$$\leq \delta \cdot \|\boldsymbol{\Psi}_u^i[j]\|_{2,N_s}. \tag{16}$$

Next, using the definition $\lambda = \sup_{s \in \mathcal{S}} \|\mathbf{Z}_u^i\|_{2,N_s}$, the second term can be bounded as

$$\left|\langle \mathbf{Z}_v^i, \boldsymbol{\Psi}_u^i[j] - \boldsymbol{\Psi}_v^i[j]\rangle_{N_s}\right| \leq \|\mathbf{Z}_v^i\|_{2,N_s} \cdot \|\boldsymbol{\Psi}_u^i[j] - \boldsymbol{\Psi}_v^i[j]\|_{2,N_s}$$
$$\leq \lambda \cdot \|\boldsymbol{\Psi}_u^i[j] - \boldsymbol{\Psi}_v^i[j]\|_{2,N_s}. \tag{17}$$

By Eq.16 and Eq.17, summing over all $j$ and $s \in \mathcal{S}$, we have that

$$\sum_{s \in \mathcal{S}} |\mathbf{Z}_u^{i+1}(s) - \mathbf{Z}_v^{i+1}(s)|^2$$
$$\leq \|\mathbf{a}^i\|_2^2 \sum_{s \in \mathcal{S}} \sum_{j=1}^{M} \cdot \left(\delta^2 \cdot \|\boldsymbol{\Psi}_u^i[j]\|_{2,N_s}^2 + \lambda^2 \cdot \|\boldsymbol{\Psi}_u^i[j] - \boldsymbol{\Psi}_v^i[j]\|_{2,N_s}^2\right). \tag{18}$$

Using $|\mathcal{S}|$ as the area of the domain of $\mathbf{Z}^i$, and summing over $s$, we can write:

$$\sum_{s \in \mathcal{S}} \sum_{j=1}^{M} \|\boldsymbol{\Psi}_u^i\|_{2,N_s}^2 = |\mathcal{S}| \cdot \|\boldsymbol{\Psi}_u^i\|_2^2, \tag{19}$$

and

$$\sum_{s \in \mathcal{S}} \sum_{j=1}^{M} \|\boldsymbol{\Psi}_u^i[j] - \boldsymbol{\Psi}_v^i[j]\|_{2,N_s}^2 = |\mathcal{S}| \cdot \|\boldsymbol{\Psi}_u^i - \boldsymbol{\Psi}_v^i\|_2^2. \tag{20}$$

Therefore, Eq.18 can be rewritten as

$$\|\mathbf{Z}_u^{i+1} - \mathbf{Z}_v^{i+1}\|_2^2 \leq \|\mathbf{a}^i\|_2^2 |\mathcal{S}| \cdot \left(\delta^2 + \lambda^2 \cdot \|\boldsymbol{\Psi}_u^i - \boldsymbol{\Psi}_v^i\|_2^2\right). \tag{21}$$

Taking the square root and applying the triangle inequality for the Euclidean norm,

$$\|\mathbf{Z}_u^{i+1} - \mathbf{Z}_v^{i+1}\|_2$$
$$\leq \|\mathbf{a}^i\|_2 \sqrt{|\mathcal{S}|} \cdot \sqrt{\delta^2 + \lambda^2 \cdot \|\boldsymbol{\Psi}_u^i - \boldsymbol{\Psi}_v^i\|_2^2}$$
$$\leq \|\mathbf{a}^i\|_2 \sqrt{|\mathcal{S}|} \cdot (\delta + \lambda \cdot \|\boldsymbol{\Psi}_u^i - \boldsymbol{\Psi}_v^i\|_2)$$
$$\leq (\|\mathbf{a}^i\|_2 \lambda \sqrt{|\mathcal{S}|} + \delta) \cdot \|\boldsymbol{\Psi}_u^i - \boldsymbol{\Psi}_v^i\|_2, \tag{22}$$

which proves that $\|\mathbf{Z}_u^{i+1} - \mathbf{Z}_v^{i+1}\|_2 \leq (\|\mathbf{a}^i\|_2 \lambda \sqrt{|\mathcal{S}|} + \delta) \cdot \|\boldsymbol{\Psi}_u^i - \boldsymbol{\Psi}_v^i\|_2$ as claimed.

# B. Experimental Details

## B.1. Implementations.

We adopt U-Net architecture from (Song et al., 2020) as our main model that predicts 3D Gaussians parameters. Each layer of the encoder and decoder parts are decomposed over a set of filter atoms and coefficients, where filter atoms are generated by Heavy Ball NODE (Xia et al., 2021) according to input camera position, i.e., azimuthal and polar angles. Heavy Ball NODE is implemented with two neural networks. First, we employed 3-layer MLP to predict the initial value and velocity (i.e., momentum) of

NODE, taking normalized polar angles as input. SiLU activation functions are used in between the three linear layers. Second, the ODE is implemented with three 1x1 convolutional layers. The starting point of ODE integration is always set to 0, while the end point is decided according to normalized azimuthal angles. Each convolutional layer is bundled with Group-Normalization and activation function. Different activation functions are used for different datasets, for stability of NODE learning. For SRN-Chairs, CO3D-Teddybears, and CO3D-Hydrants, we used Softplus function. For SRN-Cars dataset, we used Sigmoid function. For NMR multi-category ShapeNet and Objaverse, we used SiLU function. The choice of activation functions are decided empirically. For NODE learning, an adaptive step solver, Dormand-Prince method (Dopri5) (Dormand & Prince, 1980), is used. To prevent the Dopri5 solver's step size from becoming excessively small, we employed gradient clipping with max norm of the gradients set to 5.0, throughout all training.

Following (Szymanowicz et al., 2024), all models are trained in two-phases. For all datasets, models are first trained for 600k iterations with $L_2$ loss with learning rate of 5.0e-05. Then, we train additional 200k iterations with LPIPS loss. When imposing LPIPS loss in addition to $L_2$ loss, we decreased learning rate to 5.0e-06 and multiplied 0.01 and 0.99 on the LPIPS and $L_2$ loss term, respectively, to balance both losses. For rasterization, we used implementation from Gaussian Splatting (Kerbl et al., 2023).

All models are trained on a single NVIDIA RTX A5000 GPU, with full convergence taking 8 days. Our method maintains the same test-time efficiency as (Szymanowicz et al., 2024), taking 25 minutes and 14 minutes to process the whole ShapeNet-Chairs and Cars testsets, respectively. More training time comparisons with other benchmarks are presented in the Table 8.

| Method | GPU | Memory | # GPUs | Days | GPU × Days |
|---|---|---|---|---|---|
| VisionNeRF | A100 | 80G | 16 | 5 | 80 |
| NeRFDiff | A100 | 80G | 16 | 3 | 48 |
| ViewDiff | A40 | 48G | 2 | 3 | 6 |
| PixelNeRF | TiRTX | 24G | 1 | 6 | 6 |
| SplatterImage | A6000 | 48G | 1 | 7 | 7 |
| **Ours** | **A5000** | **24G** | **1** | **8** | **8** |

*Table 8.* Comparison of training resource usages.

In the Figure 9, we also present illustration of two separate view splits used for view interpolation and view extrapolation experiments presented in Figure 5.

**B.2. Datasets**

**SRN Cars and Chairs.** We use standard train, test, validation split provided in the SRN-Cars and Chairs datasets. For test time rendering, we use view 64 as the conditioning

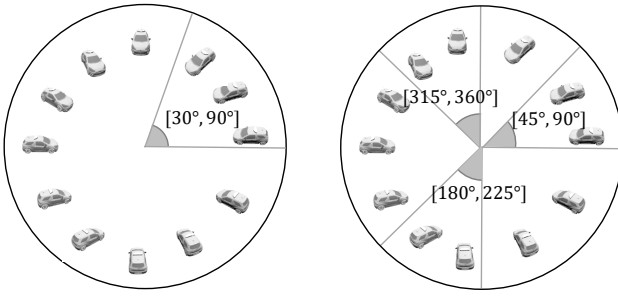

View-Space Extrapolation        View-Space Interpolation

*Figure 9.* Illustration of two separate view splits for unseen view generalization experiments. Models are trained on non-shadowed angles and tested on views from shadowed angles.

view. For generating Figure 4, we used view 5, 10, 15, 20, 25 as conditioning views when rendering novel views.

**Multi-category ShapeNet.** We use standard train, test, validation split provided from NMR (Kato et al., 2018).

**CO3D Teddybears and Hydrants.** Following (Szymanowicz et al., 2024), we used first view as the conditioning view when rendering novel views at test time. However, only sequences with first frame that has evident foreground mask with probability over 0.8 is used for object-centric reconstructions.

**Objaverse and Google Scanned Objects.** Following (Szymanowicz et al., 2024), training images from the Objaverse-LVIS (Deitke et al., 2023) are generated using the rendering pipeline from Zero-1-to-3 (Liu et al., 2023). For the testing images from the Google Scanned Objects (GSO) dataset (Downs et al., 2022), same evaluation renderings protocol adopted in Free3D (Zheng & Vedaldi, 2024) is used.

Note that except for the multi-category NMR dataset which used $64 \times 64$ resolution, all experiments are conducted at a resolution of $128 \times 128$.

## C. More Results

In this section, we show quantitative comparisons of ours and vanilla Splatter Image (Szymanowicz et al., 2024) on the ShapeNet Cars and Chairs datesets using different input images. Table 9 and Table 10 show that while the baseline, Splatter Image (Szymanowicz et al., 2024), exhibits larger inconsistencies when input views vary, our model maintains superior consistency over different datasets. This demonstrates the practical benefit of our view-adaptive approach in handling diverse input viewpoints.

Visualizations of the opacity maps generated from ours and the vanilla Splatter Image is also presented in the Figure 10.

In the remaining section, we present additional qualitative results of experiments introduced in the Section 5. To enhance visual clarity, all plots are scaled to span the full page.

| Input View | Method | ShapeNet-Cars | | | ShapeNet-Chairs | | |
|---|---|---|---|---|---|---|---|
| | | PSNR↑ | SSIM↑ | LPIPS↓ | PSNR↑ | SSIM↑ | LPIPS↓ |
| 16 | Baseline | 22.25 | 0.89 | 0.11 | 21.41 | 0.89 | **0.11** |
| | Ours | **22.43** | **0.90** | **0.09** | **21.90** | **0.90** | **0.11** |
| 32 | Baseline | 21.16 | **0.89** | 0.13 | 20.71 | 0.88 | 0.124 |
| | Ours | **21.58** | **0.89** | **0.12** | **21.23** | **0.89** | **0.122** |
| 48 | Baseline | 22.61 | 0.91 | 0.09 | 23.34 | **0.91** | **0.09** |
| | Ours | **23.12** | **0.92** | **0.08** | 22.63 | **0.91** | **0.09** |
| 64 | Baseline | 24.00 | 0.92 | 0.078 | 24.43 | **0.93** | **0.067** |
| | Ours | **24.46** | **0.93** | **0.073** | **24.76** | **0.93** | 0.071 |
| 80 | Baseline | 23.89 | 0.92 | 0.089 | 23.19 | 0.92 | 0.079 |
| | Ours | **24.27** | **0.93** | **0.072** | **24.09** | **0.93** | **0.076** |

*Table 9.* Performance comparison on ShapeNet-Cars and ShapeNet-Chairs datasets for various input views. Each number in the Input View column denotes the index of the conditioning image in the test set. PSNR, SSIM, and LPIPS are reported for both categories.

| Input View | Method | GSO | | | Teddybears | | | Hydrants | | |
|---|---|---|---|---|---|---|---|---|---|---|
| | | PSNR↑ | SSIM↑ | LPIPS↓ | PSNR↑ | SSIM↑ | LPIPS↓ | PSNR↑ | SSIM↑ | LPIPS↓ |
| 6 | Baseline | 20.21 | 0.86 | 0.12 | 18.84 | 0.70 | 0.25 | 21.79 | 0.80 | 0.15 |
| | Ours | **20.95** | **0.87** | **0.11** | **18.92** | **0.70** | **0.25** | **21.80** | **0.81** | **0.14** |
| 12 | Baseline | 20.24 | 0.86 | 0.12 | 18.56 | 0.69 | 0.26 | 21.86 | 0.80 | 0.14 |
| | Ours | **20.91** | **0.88** | **0.10** | **18.73** | **0.70** | **0.25** | **22.21** | **0.81** | **0.14** |
| 24 | Baseline | 20.28 | 0.86 | 0.12 | 18.32 | 0.68 | 0.27 | 21.82 | 0.80 | 0.13 |
| | Ours | **20.96** | **0.88** | **0.10** | **18.43** | **0.69** | **0.26** | **22.20** | **0.81** | **0.13** |

*Table 10.* Performance comparison across different conditioning viewpoints on GSO, Teddybears, and Hydrants datasets. Each number in the Input View column denotes the index of the conditioning image in the test set.

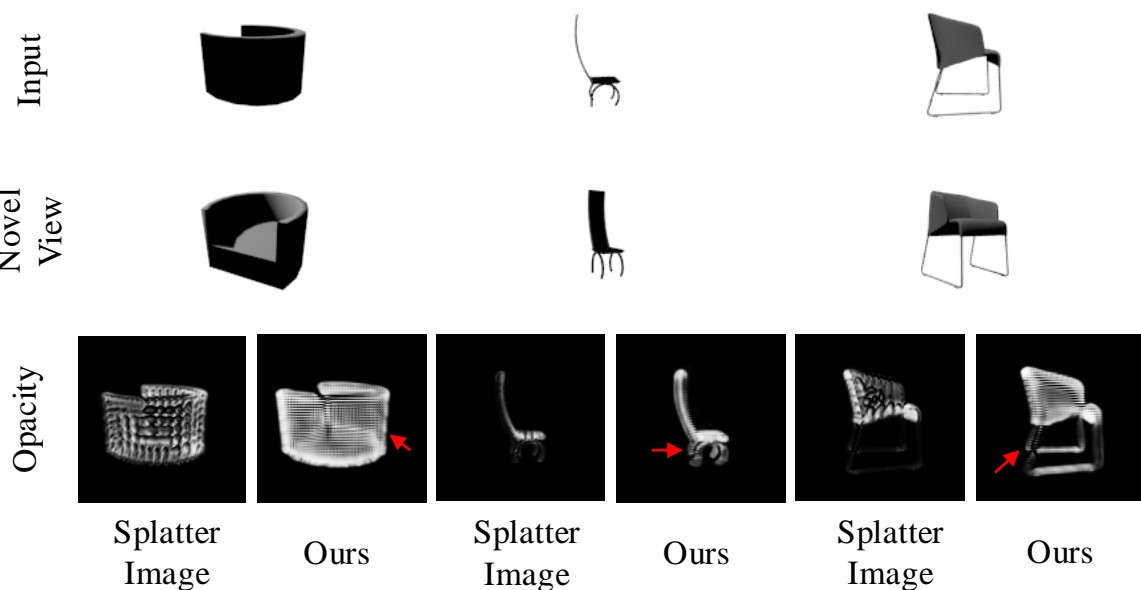

*Figure 10.* Visualizations of opacity maps generated from our method and vanilla Splatter Image.

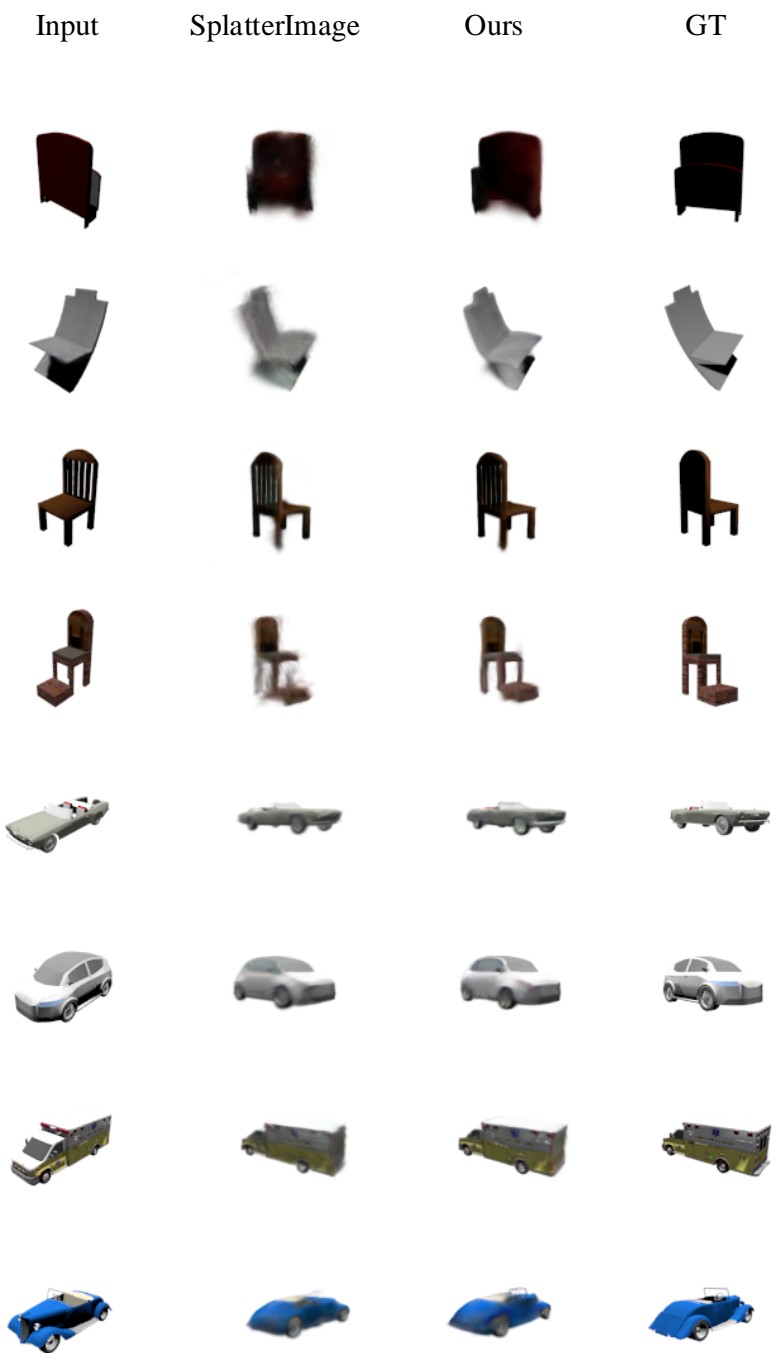

*Figure 11.* Additional visual comparison results between our approach and the Splatter Image on ShapeNet-Cars and ShapeNet-Chairs dataset.

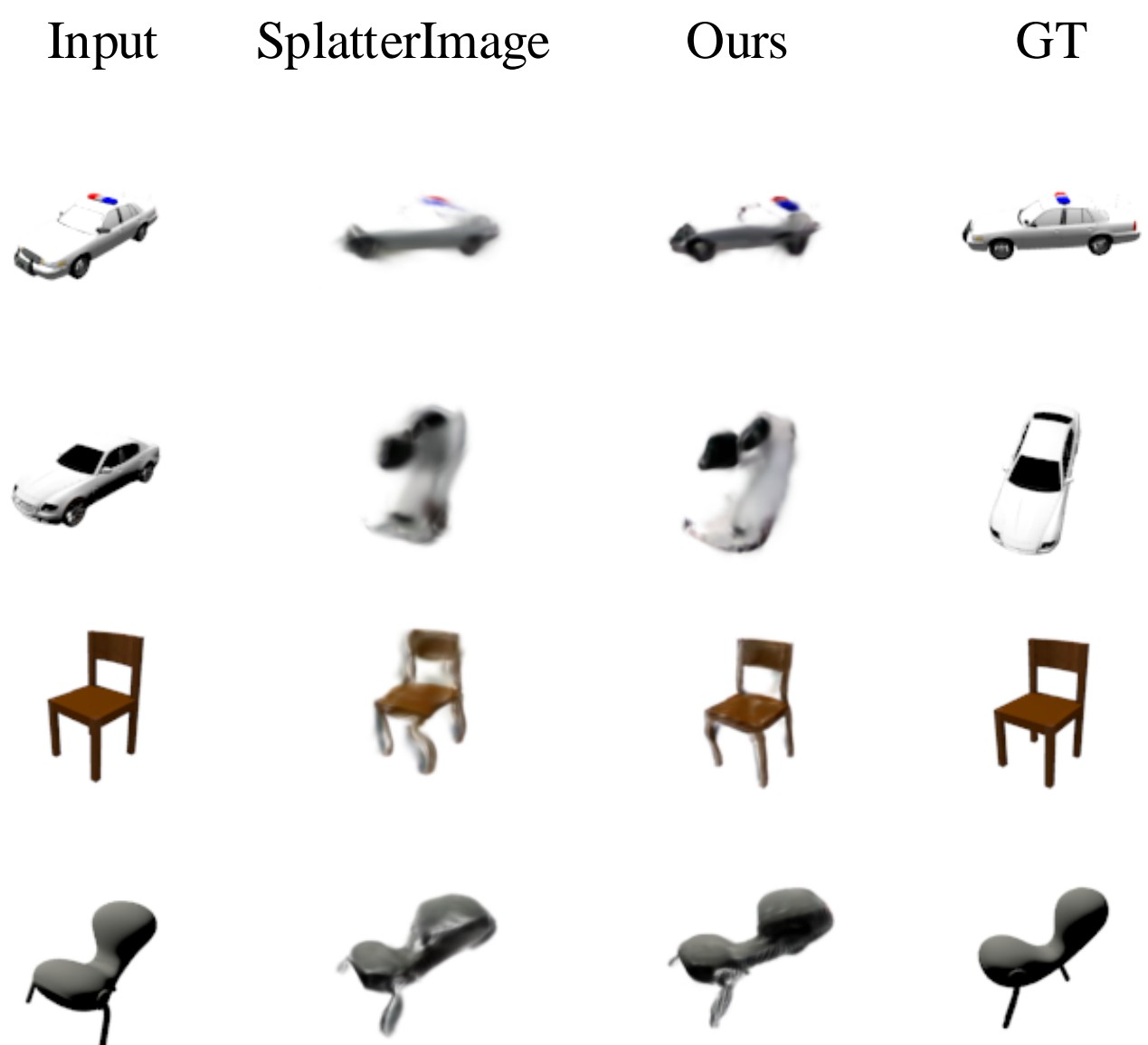

*Figure 12.* Visual comparison between our approach and the Splatter Image on unseen category generalization experiment. Models trained on ShapeNet-Cars are tested on ShapeNet-Chairs, and vice-versa. The first two rows show results of Chairs2Cars, while the last two rows show results for Cars2Chairs.

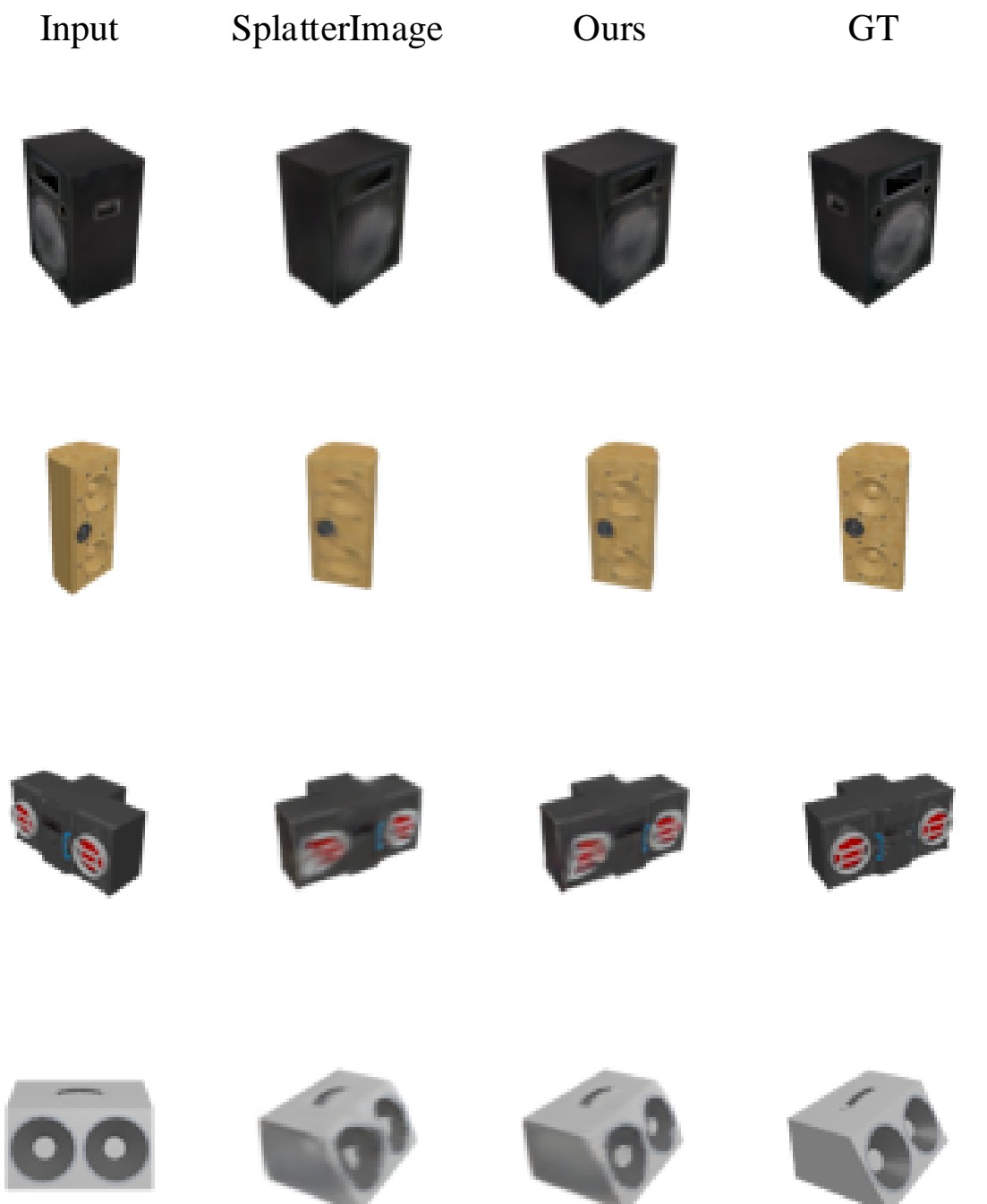

Input      SplatterImage      Ours      GT

*Figure 13.* Visual comparison between our approach and the Splatter Image on the "Speaker" category inside multi-category ShapeNet dataset.

