# OpenReview forum: "Continuous Viewpoint Adaptation for Single View 3D Object Reconstruction"
_ICML.cc/2026/Conference — ICML 2026 regular_

### Official Review · Reviewer_LuTB · 2026-02-24

**Soundness:** 3
**Presentation:** 3
**Significance:** 3
**Originality:** 3
**Overall Recommendation:** 5
**Confidence:** 5

**Summary:**

This paper proposes a novel method for single-view 3D object reconstruction, building upon the "Splatter Image" framework which predicts 3D Gaussian Splatting (3DGS) parameters via a feed-forward neural network. The authors identify a key limitation in existing feed-forward methods: sensitivity to input viewpoint changes and poor consistency when handling occlusions.

To address this, they introduce a Continuous Viewpoint Adaptation mechanism. Instead of learning static convolutional filters, the method learns a "filter subspace" where filter atoms are generated dynamically based on the input camera's polar and azimuthal angles. This generation is modeled using Neural Ordinary Differential Equations (NODE)—specifically a Heavy Ball NODE—to ensure that the evolution of network parameters is continuous with respect to camera movement. The authors provide a theoretical proposition asserting that this Lipschitz continuity in filter generation leads to bounded variations in the output 3D reconstruction, thereby enforcing view consistency. Empirical results on ShapeNet (Cars/Chairs) and CO3D datasets demonstrate that the method outperforms baselines like Splatter Image and PixelNeRF in terms of view consistency and reconstruction quality, particularly in occluded regions.

**Compliance With Llm Reviewing Policy:**

Affirmed.

**Key Questions For Authors:**

- Robustness to Pose Noise: The method relies on exact $(\Theta, \Phi)$ input. How does the performance degrade if the input camera pose is noisy or estimated? Have you experimented with perturbing the input angles during inference?

- Inference Latency: While training time is compared in Table 7, what is the impact on inference latency? Does solving the ODE to generate filter atoms for every forward pass add significant overhead compared to the static Splatter Image inference?

- ODE Solver Stiffness: Did you encounter issues with stiffness in the ODE solver during training? The paper mentions gradient clipping, but were there specific instability issues related to the diverse geometry of ShapeNet objects?

- Filter Subspace Size: How sensitive is the performance to the number of filter atoms $M$? Is there a trade-off between the size of the subspace and the smoothness of the adaptation?

**Limitations:**

The authors frankly acknowledge the primary limitation: the requirement for known camera viewpoints. This prevents direct application to unposed image collections without a preprocessing step (like pose estimation). Additionally, the method is currently limited to single objects and does not handle complex scenes with backgrounds, which is a common constraint for this class of reconstruction methods.

**Strengths And Weaknesses:**

Strengths:

- Sound Theoretical Motivation: The use of NODE to enforce continuity in the filter space is a well-motivated inductive bias for this problem. The authors successfully connect the continuous nature of camera viewpoints (spherical coordinates) to the continuous dynamics of the network weights. The theoretical proof (Proposition 4.1) regarding Lipschitz continuity provides a solid grounding for why this method should improve consistency.

- Novel Architecture Design: Decomposing the convolutional layers into view-shared coefficients and view-dependent filter atoms generated by NODE is an elegant way to condition the network on geometric priors without exploding the parameter count. This effectively "embeds" the perspective distortion rules into the network structure.

- Improved View Consistency: The qualitative results (e.g., Figure 4 and Figure 9) convincingly show that the proposed method produces more stable reconstructions across varying input views compared to the baseline Splatter Image, which suffers from "flickering" or shape shifting when the input view changes slightly.

- Generalization: The experiments on unseen categories (Table 5) and view extrapolation (Table 4) suggest that the learned continuous dynamics help the model generalize better than static feed-forward networks.

Weaknesses:

- Reliance on Accurate Camera Poses: The method explicitly conditions the network on input camera angles $(\Theta, \Phi)$. While this is standard for benchmarks like ShapeNet, it limits the application in "in-the-wild" scenarios where camera poses might be unknown or noisy. The paper mentions this in the limitations but does not provide experiments on how robust the model is to noisy pose inputs.

- Marginal Quantitative Gains on Real Data: While the method shows clear improvements on synthetic ShapeNet data, the quantitative gains on the real-world CO3D dataset (Table 3) are relatively marginal compared to the baseline Splatter Image (e.g., roughly +0.06 PSNR for Teddybears, +0.01 for Hydrants). The visual improvements in consistency seem more significant than the reconstruction metrics suggest.

- Training Complexity: The introduction of NODE and the ODE solver (Dopri5) likely introduces training overhead compared to a standard U-Net. Table 7 confirms that training takes 8 days compared to 7 for Splatter Image, which is a manageable increase, but the complexity of implementation is higher.

---

> ### Author Rebuttal · Authors · 2026-03-31
>
> ### Robustness to input angle perturbations.
> We evaluate robustness to pose noise by perturbing both azimuth and polar angles at inference time. Specifically, we introduce noise levels of 5 percent, 10 percent, and 15 percent to the input camera pose. On the Teddybears dataset, PSNR changes from 18.55 to 18.04 and 17.74 as noise increases. On Hydrants, PSNR changes from 21.62 to 21.57 and 21.28 under the same perturbations. These results show that the model remains stable under moderate pose noise, with gradual degradation as noise increases. This indicates that the method is reasonably robust to imperfect camera pose inputs while maintaining consistent reconstruction quality.
>
> ---
>
> ### Marginal quantitative gains on real data:
> For further evaluation on real-world data, we include large scale real data evaluation using Objaverse for training and Google Scanned Objects (GSO) for testing, which covers over 1k object categories. Please refer to the Table 9 and the qualitative figure, presented under response to Reviewer BPno for the results. This setting demonstrates clear improvements across all metrics over the baseline, under diverse real world categories.
>
> In addition, to compensate the marginal performance gains on CO3D, we further analyze performance across different conditioning views, which provides a more comprehensive assessment of robustness in single view 3D reconstrucion problem. When evaluated under varying input viewpoints, our method consistently achieves better performance than the baseline across all views on datasets such as GSO, CO3D Teddybears and Hydrants. Please refer to Table 10 from the response under Reviewer 7qBD for the results. The results show that improvements are consistent across different input views, rather than concentrated in a single evaluation setting, although it might appear marginal for some input views. This per view consistency directly reflects improved robustness to conditioning images, which is a central goal of our method.
>
> ---
>
> ### Training Complexity
> The reported training time reflects our current resource setting rather than an inherent limitation of the method. Our model is trained on a single NVIDIA A5000 GPU, while prior work such as Splatter Image uses higher end hardware such as NVIDIA A6000. With access to multiple GPUs and larger batch sizes, the training time can be further reduced. In terms of implementation, the proposed convolutional filter decomposition and Neural ODE module are lightweight additions that operate at the level of convolutional weights. They can be integrated into existing convolutional architectures in a plug-and-play manner without requiring substantial changes to the overall network design. We will release the code upon acceptance to facilitate reproducibility and ease of adoption.
>
> ---
>
> ### Inference Latency
>
> The proposed method introduces minimal overhead at inference time. Although filter atoms are generated through ODE integration, this computation is lightweight and does not significantly affect runtime. We report the comparable average inference time per instance over ShapeNet Chairs samples. Please refer to the Table 11 under response to Reviewer BPno for the results.
>
> ---
>
> ### ODE Solver Stiffness
> During training, we did not observe stiffness related issues on ShapeNet. For CO3D, we occasionally encountered solver instability such as excessively small step sizes with the Dopri solver after extended training. This was effectively addressed through the choice of activation functions used in the Neural ODE. In particular, using smoother activations which are less prone to vanishing gradients, e.g., SiLU or SoftPlus, rather than ReLU or Sigmoid functions stabilized the ODE dynamics. After this adjustment, training proceeded reliably without further instability. We attribute this behavior to the increased complexity and variability of real world data in CO3D, which requires additional care in modeling the ODE function.
>
> ---
>
> ### Filter Subspace Size
> The number of filter atoms is chosen empirically following prior work on decomposed convolution such as DCF-Net, where a moderate number of basis filters is sufficient to capture the filter space. In practice, varying the number of atoms does not cause instability, but changes the capacity of the subspace, similar to standard basis expansion.
>
> The smoothness of viewpoint adaptation is governed by the continuous parameterization of filter atoms through the Neural ODE, rather than the number of atoms. Smooth transitions are enforced by modeling filter atoms as a function of camera pose through ODE integration over polar and azimuth angles, ensuring nearby viewpoints produce smoothly varying filters.
>
> The number of atoms controls expressiveness, while the ODE defines continuous evolution across viewpoints. Thus, subspace size and smoothness play complementary roles, and changing the number of atoms does not affect continuity.

---

> > ### Author Rebuttal · Reviewer_LuTB · 2026-04-05
> >
> > No further comments.

---

### Official Review · Reviewer_XUe5 · 2026-03-02

**Soundness:** 2
**Presentation:** 3
**Significance:** 2
**Originality:** 2
**Overall Recommendation:** 4
**Confidence:** 3

**Summary:**

This paper proposes a novel approach for single-view 3D object reconstruction using 3D Gaussian Splatting, enhanced by a continuous viewpoint adaptation mechanism. The key innovation is modeling the filter subspace of a neural network with Neural Ordinary Differential Equations (NODE)  to handle the continuity of camera viewpoints, represented by polar and azimuthal angles. Experiments claim superior performance on benchmarks and better generalization to unseen angles and categories.

**Compliance With Llm Reviewing Policy:**

Affirmed.

**Final Justification:**

I have updated my score from 3 to 4 after reading the rebuttal. The authors addressed the main issues I raised and clarified the intended scope of the paper more clearly. In particular, their response helped explain that the main value of the method is in improved robustness and generalization across views, rather than uniform gains on every single metric. The additional results and clarifications also make the empirical evidence more convincing, especially regarding the ablation setting and efficiency comparison.

I still have some reservations. The use of NODE for continuous filter modeling still feels more like an extension of existing ideas than a fundamentally new contribution, and some of the performance gains are still relatively modest. So I do not see this as a particularly strong accept. However, the rebuttal does resolve enough of my earlier concerns that I no longer think rejection is warranted. For that reason, I am raising my score to 4.

**Key Questions For Authors:**

1.Why are ablation studies conducted with shorter training iterations? How do results change with full training, and does this affect claims of component effectiveness?

2.On datasets like ShapeNet-Chairs and CO3D, improvements are small or inconsistent (e.g., worse LPIPS in some cases). Can you provide statistical significance tests or more diverse benchmarks to justify "state-of-the-art"?

3.Training takes 8 GPU-days—longer than baselines like Splatter Image and PixelNeRF.

**Limitations:**

The method assumes accurate camera pose estimation, but robustness to pose noise is not evaluated. While presented as an improvement over Splatter Image for better view consistency, the work offers only incremental practical gains. Algorithmic novelty and theoretical contributions to continuous representations or view-invariant learning remain limited and underexplored.

**Strengths And Weaknesses:**

**strength**

1.The authors propose a continuous view-adaptive model to improve consistency across different viewpoints, addressing the single-view 3D object reconstruction problem.

2.This work addresses the 3D reconstruction problem of handling viewpoint changes and occlusions, proposing an innovative solution using NODE to model filter subspaces, effectively capturing the continuity of camera rotations.

3.The authors demonstrate the robustness and performance of their method across various benchmark datasets, particularly excelling in generalization to unseen viewpoints and categories.

**weakness**

1.The technical foundation draws from established works like Gaussian Splatting and NODE, but the claims of superior performance are not fully substantiated.  For instance, while the PSNR improvement of 0.33 dB on ShapeNet-Chairs might be considered marginal, the fact that LPIPS slightly worsens (0.071 vs. 0.067) raises questions about the consistency of the improvement. A statistical significance test across multiple runs would help clarify whether these differences are meaningful or simply within the noise floor. The ablation studies use shorter training iterations, making direct comparisons to full models unfair and potentially inflating perceived gains from components like NODE conditioning.

2.The paper is structured logically, with clear sections on methods and experiments.  The narrative claims "state-of-the-art" performance, yet tables show only incremental gains.

3.The problem of viewpoint sensitivity in single-view reconstruction is relevant, but the contributions offer limited advancement. Generalization to unseen views/categories is tested, but results are modest, and the method is restricted to single objects. It is unclear how well the method would extend to more complex scenes.

4.The integration of NODE for continuous filter modeling is a reasonable extension, but not highly novel—it combines existing ideas without deep new insights. Distinctions from Splatter Image are clear but incremental, focusing on view adaptation via ODEs.

---

> ### Author Rebuttal · Authors · 2026-03-31
>
> ### Superior performance and statistical significance test
> We thank the reviewer for the insightful comments. We clarify that our primary claim is not uniform improvement on every metric such as Chairs LPIPS, but consistent gains in robustness and generalization, which are more indicative for single image 3D reconstruction.
>
> To further demonstrate scalability and generalization, we conduct experiments on the large scale Objaverse-LVIS dataset, which contains over 1k object categories. When evaluated on Google Scanned Objects (GSO), our method consistently outperforms the baseline across all conditioning views, indicating strong robustness and generalization under diverse categories and inputs, as shown in the first column of Table 10 under response to Reviewer 7qBD. Please also refer to Table 9 and qualitative figure presented under response to Reviewer BPno, for evaluation results on GSO dataset under fixed input view.
>
> For statistical significance, due to short rebuttal perior, instead of relying only on multiple training seeds, we evaluate consistency across different conditioning views. In single image 3D reconstruction, the random choice of input view strongly affects reconstruction difficulty. Using different conditioning views naturally introduces variation. Thus, we present statistical significance through multiple runs of 3D reconstruction tasks under different conditioning views.
>
> We explicitly report these results in Table 8 in the Appendix and Table 10 from the response under Reviewer 7qBD, where each entry corresponds to a reconstruction scores resulting from different conditioning views. Both table demonstrates that our method consistently outperforms the baseline across most views and metrics across ShapeNet, CO3D, and GSO datasets, showing that improvements are not due to randomness from a particular input. When averaging PSNR, LPIPS, and SSIM scores for different conditioning views, our model outperformed the baseline for all metrics across all datasets.
>
> ---
>
> ### Shorter iterations on ablation study
> The ablation study is not intended for direct comparison with fully trained models, but to isolate the contribution of each component. All ablation variants are trained under the same shortened schedule, ensuring fair comparison among them. The lower absolute performance compared to the main results is expected due to reduced training. This follows prior work such as Splatter Image, which also adopts shorter training schedules for ablation studies.
>
> ---
>
> ### Extension to multi objects and complex scenes reconstruction problem
> To demonstrate scalability and generalization, we train on Objaverse evaluate on GSO. Both datasets include scenes with multiple objects. Our method consistently outperforms the baseline under these diverse compositions. For example, our model shows stronger reconstruction quality on the "Construction Set" across input views compared to Splatter Image, as shown in the first column of the attached visual results [link](https://shorturl.at/A0JtH). This indicates that although the method targets single view reconstruction, it has potential to extend to multi object scenarios.
>
> While experiments focus on single objects for standardized evaluation, the formulation is not inherently restricted. The core idea of continuous viewpoint conditioned filter generation can extend to more complex scenes where viewpoint variation remains important. We consider extending to such settings as future work.
>
> ---
>
> ### Novelty
>
> To the best of our knowledge, prior works do not embed continuous viewpoint variation directly into model weights. Existing methods, including Splatter Image, condition on camera parameters at the input or feature level while keeping weights fixed. In contrast, our method models network parameters as a continuous function of camera pose, enabling smooth adaptation across viewpoints. Neural ODEs are used to parameterize this transformation in a principled way, capturing continuity of camera motion not enforced in prior designs.
>
> We also identify viewpoint sensitivity as a key challenge in single image 3D reconstruction. Our formulation improves robustness to input views by incorporating this inductive bias into the model, offering a new direction for enhancing viewpoint consistency.
>
> ---
>
> ### Training time
> The reported training time reflects our current resource setting rather than an inherent limitation of the method. Our model is trained on a single NVIDIA A5000 GPU, while prior work such as Splatter Image uses higher-end hardware, i.e., NVIDIA A6000 GPU. With access to multiple GPUs and larger batch sizes, the training time can be significantly reduced. Importantly, the proposed method does not introduce much overhead at inference time. Please refer to the Table 11 presented in the response under Reviewer BPno for inference time comparison.
>
> ---
>
> ### Robustness to input angle perturbations.
> Please refer to the response submitted to Reviewer LuTB under same subtitle.

---

> > ### Author Rebuttal · Reviewer_XUe5 · 2026-04-02
> >
> > I appreciate the authors’ rebuttal and the additional experiments.  The response addresses the main concerns I raised and helps clarify both the scope of the paper and the intended contribution.  I still suspect that the use of NODE for continuous filter modeling is more a reasonable extension of existing ideas than a particularly novel contribution. I also think some of the original claims should be stated a bit more carefully, given that the gains on several benchmarks are still fairly modest.  That said, the rebuttal does strengthen the paper and increases my confidence in the overall results.  I will change the score of my review to 4.

---

> > > ### Author Response · Authors · 2026-04-03
> > >
> > > Thank you for the thoughtful feedback and for revisiting your assessment after our rebuttal. We truly appreciate your time and consideration. We will further refine the framing of our contributions and claims to better reflect the scope and empirical gains in the final version.

---

### Official Review · Reviewer_BPno · 2026-03-03

**Soundness:** 2
**Presentation:** 3
**Significance:** 3
**Originality:** 3
**Overall Recommendation:** 4
**Confidence:** 5

**Summary:**

Building upon the Splatter Image, this paper adopts Neural Ordinary Differential Equations(NODE) to address inherent continuity of camera viewpoints.  The convolutional filter of the base U-Net is decomposed over a set of M filter atoms and coefficient. To capture the continuous nature of spherical camera coordinates, the authors employ NODE to generate view-conditional filter atoms. The initial
conditions for position and momentum are learned functions of the polar angle. Smooth View Adaptation and View Consistency ensures that minor input perturbations lead to smooth, bounded changes in the reconstructed 3D shape. The approach show state-of-the-art performance on several single-view 3D object reconstruction benchmarks.

**Compliance With Llm Reviewing Policy:**

Affirmed.

**Final Justification:**

The rebuttal and replies address most of my concerns, and I will maintain the positive rating.

**Key Questions For Authors:**

Is this method transferable to non-convolutional architectures?

**Limitations:**

yes

**Strengths And Weaknesses:**

Strengths:

1 Decomposing convolutional filters through NODE to obtain view-adaptive filter atoms is a good idea for handling viewpoint issues.

2 The method is supported by detailed mathematical reasoning and justification.

3 The ablation experiments clearly demonstrate the role played by each design component of the method.

Weaknesses:

1 Some single-view 3d reconstruction methods based on Gaussian Splatting, such as Gamba[1] and Triplane Meets Gaussian Splatting[2], were not included in the comparison.

2 The comparison of testing efficiency between different methods should be conducted on the same GPU.

3 No experiments were conducted on the Objaverse-LVIS and Google Scanned Objects datasets.

4 This method is based on convolutional filter and cannot be applied to non-convolutional architectures.

5 The need to use different activation functions for different datasets reduces the generalizability of this method.

[1] Gamba: Marry Gaussian Splatting with Mamba for Single-View 3D Reconstruction.
[2] Triplane Meets Gaussian Splatting: Fast and Generalizable Single-View 3D Reconstruction with Transformers.

---

> ### Author Rebuttal · Authors · 2026-03-31
>
> ### Additional comparisons
>
> The suggested methods Gamba[1] and Triplane Meets Gaussian Splatting[2] rely on pre-trained ViT based encoders such as DINO or DINOv2 to extract patch level features, and adopt transformer based architectures for prediction. In contrast, our method is designed for convolutional networks and does not rely on large scale pre-trained models, i.e., our convolutional encoder is trained from scratch. Due to these architectural differences, directly applying our method to these transformer based pipelines is not straightforward, and the comparison would not be fully aligned in terms of model design and assumptions. However, we will include and discuss these works in the related work section to clarify their differences in scope.
>
> ---
>
> ### Testing efficiency
>
> Both methods are evaluated on the same hardware for a fair comparison, using a single NVIDIA A5000 GPU. We measure inference time over 1,317 ShapeNet Chairs samples and report the average per instance:
>
> **Table 11: Inference Time**
> | Method          | Inference Time per Instance |
> |-----------------|-----------------------------|
> | Splatter Image  | 0.0291 s                    |
> | Ours            | 0.0322 s                    |
>
> The results show that our method maintains comparable inference efficiency under identical evaluation settings.
>
> ---
>
> ### Validation on Objaverse and GSO
>
> To further demonstrate scalability and generalization, we conduct experiments on the large scale Objaverse LVIS dataset, which contains over 1k object categories. When evaluated on Google Scanned Objects, our method consistently outperforms the baseline across all conditioning views, indicating strong robustness and generalization under diverse categories and inputs.
>
> **Table 9: Experiment on Google Scanned Objects**
> | Method            | PSNR ↑ | SSIM ↑ | LPIPS ↓ |
> |------------------|--------|--------|--------|
> | OpenLRM          | 18.06  | 0.84   | 0.129  |
> | Splatter Image   | 21.06  | 0.88   | 0.110  |
> | **Ours**         | **21.74** | **0.89** | **0.09** |
>
> Visual results can be found in the provided anonymous link [link](https://shorturl.at/A0JtH).
>
> ---
>
> ### Application to non-convolutional architectures
>
> The current formulation is based on decomposition of convolutional filters, and is therefore designed for convolutional architectures.
> That said, the underlying idea of parameterizing model components as continuous functions of viewpoint can potentially be extended beyond convolutional layers. For example, a similar decomposition could be applied to attention mechanisms of transformers by modeling projection matrices or attention weights as functions of camera pose, enabling view adaptive behavior in transformer based architectures. Exploring such extensions requires additional design considerations and is left as a promising direction for future work.
>
> ---
>
> ### Need of different activation functions for different datasets
>
> The choice of activation functions is empirical and not a hard requirement of the proposed method. Different activations such as SiLU, SoftPlus, or Sigmoid were selected based on performance tuning, but our method itself does not depend on a specific activation. In practice, a single activation function can be used across datasets without affecting the formulation. Additionally, activation functions are independent of learned weights and can be easily adjusted at inference time without impacting the trained parameters.

---

> > ### Author Rebuttal · Reviewer_BPno · 2026-04-07
> >
> > 1 No results on Objaverse-LVIS.
> >
> > 2  Anonymous link is unreachable.
> >
> > 3 Is the worst activation function also better than the comparative methods?
> >
> > 4 Can this method also be implemented by using the pre-trained ViT first?
> >
> >
> > Thank you for the rebuttal. It addresses most of my concerns, and I will maintain the positive rating.

---

> > > ### Author Response · Authors · 2026-04-07
> > >
> > > ### We thank the reviewer for the helpful questions and clarifications.
> > > ---
> > > 1. We would like to further clarify that the Table 9 reports results for a model trained on Objaverse-LVIS and evaluated on the external, unseen Google Scanned Objects (GSO) dataset. This protocol follows the standard evaluation setting used in Splatter Image, which we adopt to ensure fair and direct comparison. We also note that the Objaverse-LVIS does not provide a standardized test split, making it difficult to conduct consistent and reproducible evaluations across different works. As a result, it is common practice to evaluate models trained on Objaverse-LVIS using the GSO dataset. For instance, the Gamba[1] and Triplane Meets Gaussian Splatting[2] works also follow the same setting. Our setup follows this established protocol, enabling standardized and comparable benchmarking while also demonstrating generalization to unseen real-world objects.
> > > ---
> > > 2. We kindly suggest retrying the anonymous link. For convenience, we provide the link again below. As a reminder, the link contains qualitative results for a model trained on Objaverse-LVIS and tested on the GSO dataset. The first row shows the input single-view image, the second and third rows present novel view renderings from Splatter Image and our method, respectively, and the final row shows the ground truth (GT) novel views. We plan to add this figure in the Appendix of our revised manuscript. The link :
> > > [Please Click Here](https://drive.google.com/file/d/1Pl38E9hrTTfJJ_Lej5VZJusf-fHwseZt/view?usp=sharing)
> > > ---
> > > 3. **(Case 1 – Changing the activation function after training)**
> > > Changing the activation function after training degrades performance and can lead to results worse than the comparative baselines. The activation function is an integral part of the model definition during training and must remain consistent at inference time in convolutional neural networks. Modifying it post-training alters the learned feature transformations, inevitably causing performance degradation. All results reported in the experiments use consistent activation functions between training and inference.
> > >
> > > **(Case 2 – Different activation choices during training)**
> > > When the same activation function is consistently used during both training and testing, the choice of activation does influence the final model performance, but the impact is relatively modest and does not affect the overall effectiveness of the method. For example, on the Hydrants dataset, we report the results in Table 12. A model trained and evaluated with Softplus still outperforms Splatter Image and PixelNeRF in PSNR and SSIM, while remaining competitive in LPIPS. Our main results use SiLU, which achieves the best overall performance and is therefore reported in Table 3 of the main manuscript.
> > >
> > > These results indicate that the choice of activation function primarily affects optimization quality rather than the validity of the method. Selecting the best-performing activation is a standard hyperparameter tuning step and an essential part of training modern deep learning models.
> > >
> > > **Table 12: Experiment with different activation functions**
> > >
> > > | Method            | Activation | PSNR ↑ | SSIM ↑ | LPIPS ↓ |
> > > |-------------------|------------|--------|--------|--------|
> > > | PixelNeRF         | -          | 21.76  | 0.78   | 0.207  |
> > > | Splatter Image    | -          | 21.80  | 0.80   | 0.150  |
> > > | Ours              | Softplus   | **21.84**  | 0.81   | 0.150  |
> > > | **Ours (main)**   | SiLU       | 21.81  | **0.82**   | **0.147**  |
> > > ---
> > > 4. The core idea of our method is to train an encoder–decoder style convolutional network from scratch, where the encoder itself learns to extract view-specific features through the proposed decomposed filter atoms. In particular, the Neural ODE generates filter atoms that adapt continuously with respect to the input viewpoint, enabling the encoder to produce viewpoint-aware representations during feature extraction. Using a pre-trained ViT as a front-end would not align with this design principle, as it would rely on externally learned features rather than allowing our model to learn viewpoint-conditioned representations end-to-end. This would bypass the key mechanism of our approach, where viewpoint information is embedded directly into the network parameters during encoding. Therefore, incorporating a pre-trained ViT is unfortunately not consistent with the current implementation.

---

### Official Review · Reviewer_7qBD · 2026-03-12

**Soundness:** 3
**Presentation:** 2
**Significance:** 3
**Originality:** 3
**Overall Recommendation:** 4
**Confidence:** 3

**Summary:**

The authors address the problem of single-view 3D object reconstruction and highlight that the reconstruction quality of prior work is sensitive to the selected camera viewpoints. To account for this, the authors parametrize their model using Neural Ordinary Differential Equations (NODE); thereby, continuously modeling the dynamic evolution of varying camera viewpoints. As applying NODE to the full space of convolutional filters is computationally infeasible, the authors adopt a filter-subspace approach. The authors evaluated their method on the ShapeNet dataset and category subsets of the ShapeNet-SRN and CO3D datasets.

**Compliance With Llm Reviewing Policy:**

Affirmed.

**Final Justification:**

After the authors' rebuttal, I confirm the positive score. This is aligned with the other reviewers.

**Key Questions For Authors:**

Could the authors’ approach also be extended to accommodate multiple input viewpoints, as in Splatter Image?

**Limitations:**

yes

**Strengths And Weaknesses:**

Strengths:
- The paper appears technically sound (soundness). The authors highlight the strengths and limitations of their approach accordingly.
- Their use of NODE for this problem is interesting and novel (originality).
- The submitted paper is well written and well structured (presentation). The paper motivates the problem of sensitivity regarding the reconstruction quality when selecting the input view. The overall narrative is mostly easy to follow.

Weaknesses:
- **W1: Claim validation limited to one object class (soundness).** The authors should validate their claim of reduced sensitivity to input variations across additional categories beyond “Chair”.
- **W2: Limited qualitative results (soundness).** It would be helpful if the authors provided videos (in the supplementary) of the presented objects (rotating through different polar and azimuth angles) to further support their claim of improved multi-view consistency in 3D reconstructions. Evaluating 3D artifacts using rendering metrics or a small subset of smoothly interpolated camera views (Figure 4) is difficult. For example, Gaussians may render well from certain views while showing clear 3D inconsistencies from others.
- **W3: Limited applicability (significance).** The method appears tightly coupled with Splatter Image, which may limit its general applicability.
- **W4: Statement that could benefit from visual support (presentation).** The authors state that “reconstruction quality is highly sensitive to the choice of input image, as changes in viewpoint can lead to substantial variations in the output geometry.” => It would be great if the authors could show this qualitatively, e.g., in a motivational figure with Splatter Image and optionally Zero123; this would qualitatively show that the problem is inherent to this domain.

Minor:
- At the top of pages other than the first one, the “Submission and Formatting Instructions for ICML 2026” title should be replaced by the actual paper title.
- In Table 2, it would be more intuitive to switch methods and metrics (as for Table 1).

---

> ### Author Rebuttal · Authors · 2026-03-31
>
> ### Further validation for input variations sensitivity
>
> To further validate robustness to input variations beyond a single object class, we conduct experiments on the large scale Objaverse dataset, which contains over 1k object categories. The model is trained on Objaverse and evaluated on Google Scanned Objects (GSO), covering diverse categories and structures. We additionally analyze performance across multiple conditioning views on CO3D Hydrants and Teddybears datasets to assess sensitivity to input variation.
>
> **Table 10: Performance across conditioning views**
>
> | Input Viewpoint Index | Method | GSO (PSNR/SSIM/LPIPS) | Teddybears | Hydrants |
> |------|--------|------------------------|------------|----------|
> | 6    | Baseline | 20.21 / 0.86 / 0.12 | 18.84 / **0.70** / **0.25** | 21.79 / 0.80 / 0.15 |
> |      | **Ours** | **0.95 / 0.87 / 0.11** | **18.92 / 0.70 / 0.25** | **21.80 / 0.81 / 0.14** |
> | 12   | Baseline | 20.24 / 0.86 / 0.12 | 18.56 / 0.69 / 0.26 | 21.86 / 0.80 / **0.14** |
> |      | **Ours** | **20.91 / 0.88 / 0.10** | **18.73 / 0.70 / 0.25** | **22.21 / 0.81 / 0.14** |
> | 24   | Baseline | 20.28 / 0.86 / 0.12 | 18.32 / 0.68 / 0.27 | 21.82 / 0.80 / **0.13** |
> |      | **Ours** | **20.96 / 0.88 / 0.10** | **18.43 / 0.69 / 0.26** | **22.20 / 0.81 / 0.13** |
>
> Across all datasets and conditioning views, the proposed method consistently improves reconstruction quality. This demonstrates that the observed gains are stable across different categories and input viewpoints, rather than being specific to a single object class or particular input condition.
>
> ---
>
> ### Limited qualitative results
> To provide more comprehensive qualitative evaluation, we include video results of reconstructed objects in the provided anonymous links where the videos present rotations for datasets such as Hydrants [link](https://shorturl.at/ICM6v) and Teddybears [link](https://shorturl.at/vdxU1). For each example, we show ground truth, Splatter Image, and our method under identical rendering trajectories. This allows direct comparison of multi view consistency and geometric coherence across all viewing directions, rather than a limited subset of views. These results are intended to provide a clearer assessment of 3D consistency across viewpoints. Videos will be included in Appendix through respective links.
>
> ---
>
> ### Limited applicability
> While our implementation is built on top of Splatter Image, the proposed method itself is not specific to this architecture. The core idea of decomposing convolutional filters into view adaptive atoms can be applied to general convolutional neural networks used for 3D reconstruction. The formulation operates at the level of convolutional weights and can be integrated in a plug-and-play manner without requiring changes to the overall network design. This makes the approach broadly applicable beyond the specific baseline used in this work. We plan to release the code upon acceptance to facilitate adoption and further exploration in other architectures and settings.
>
> ---
>
> ### Presentation of input view sensitivity
> The sensitivity of reconstruction quality to input viewpoint is illustrated in Figure 4 of the main manuscript, which serves as a motivational example highlighting this inherent limitation. The first row in the Figure 4 shows conditioning images with small changes in viewpoint, where each column corresponds to a 15 degree rotation. The subsequent rows present rendered novel views from our method and the baseline. The comparison shows that small variations in the input view lead to noticeable degradations in reconstruction quality for the baseline, particularly in occluded regions, while our method produces more consistent results across viewpoints.
>
> In addition, this effect is supported quantitatively in Table 8 in Appendix and Table 9 under the response to Reviewer BPno, where reconstruction performance is reported across different conditioning views on multiple datasets such as ShapeNet, CO3D, and Google Scanned Objects. These results show that reconstruction quality varies significantly with input viewpoint, confirming that this sensitivity is an inherent characteristic of the task.
>
> ---
>
> ### Extension to multiple input viewpoints
> The current formulation focuses on the single input viewpoint setting, where the view adaptive filter atoms are generated through ODE integration from a single initial condition, i.e., conditioning pose. That said, the framework can be extended to multiple input viewpoints by incorporating multiple conditioning poses into the initialization or by aggregating multiple ODE trajectories. This would allow the model to leverage additional geometric cues from multiple views while preserving the continuous viewpoint modeling. We consider extending the method to multi view settings as a natural direction for future work.
>
> ---
>
> ### Minor fix
> Thanks for the correction. We will replace the dummy title with the actual paper title and correct the table accordingly.

---

> > ### Author Rebuttal · Reviewer_7qBD · 2026-04-02
> >
> > Thank you for providing the rebuttal. I consider my concerns sufficiently addressed.
> >
> > A couple of notes:
> > - The PSNR on GSO for "Ours" with Input Viewpoint Index 6 is reported as 0.95 and marked bold. I suppose that was 20.95?
> > - There is a reference to Table 9 responding to Reviewer BPno, but that was probably Table 10 in response to me.
> > - Not sure the external links with qualitatives are ok for the rebuttal, but I appreciate the effort.

---

> > > ### Author Response · Authors · 2026-04-03
> > >
> > > We very much appreciate your timely response and appreciation!
> > >
> > > To answer your notes,
> > >
> > > * The 0.95 score of PSNR marked as bold on GSO for "Ours" in the Table 10 is a typo. It should be corrected to 20.95.
> > > * In your response, we wrongly referenced Table 9. We meant to refer to the Table 10 in your response. Sorry for the confusion.
> > > * The external links provided are completely anonymous, which we believe ok for the rebuttal since it complies with the double-blind policy.

---

### Decision · Program_Chairs · 2026-04-30

**Decision:**

Accept (regular)

**Comment:**

This paper receives 3x weak accepts and 1x accept. All reviewers agree that the proposed method is technically sound, the use of NODE for this problem is interesting and novel, the method is supported by detailed mathematical reasoning and justification. The architectural design is novel, and the experiments supports the effectiveness of the proposed method. The AC follows the suggestions of all reviewers to accept the paper.